# Multi-model simulations of a springtime dust storm over Northeastern China: Implications of an evaluation of four commonly used air quality models (CMAQ v5.2.1, CAMx v6.50, CHIMERE v2017r4, and WRF-Chem v3.9.1)

Siqi Ma[1,2], Xuelei Zhang[1,3], Chao Gao[1,2], Daniel Q. Tong[3], Aijun Xiu[1], Guangjian Wu[4,5], Xinyuan Cao[1,2], Ling Huang[6], Hongmei Zhao[1], Shichun Zhang[1,7], Sergio Ibarra-Espinosa[1,8], Xin Wang[9], Xiaolan Li[10,11], and Mo Dan[12]

[1]Key Laboratory of Wetland Ecology and Environment, Northeast Institute of Geography and Agroecology, Chinese Academy of Sciences, Changchun 130102, China
[2]University of Chinese Academy of Sciences, Beijing 100049, China
[3]Center for Spatial Information Science and Systems, George Mason University, Fairfax, VA 22030, USA
[4]Key Laboratory of Tibetan Environment Changes and Land Surface Processes, Institute of Tibetan Plateau Research, Chinese Academy of Sciences, Beijing 100101, China
[5]CAS Center for Excellence in Tibetan Plateau Earth Sciences, Beijing 100101, China
[6] School of Environmental and Chemical Engineering, Shanghai University, Shanghai 200444, China
[7]Department of Marine, Earth, and Atmospheric Sciences, North Carolina State University, Raleigh, NC 27695, USA
[8]Department of Atmospheric Sciences, Universidade de S ão Paulo, S ão Paulo, SP, Brazil
[9]Key Laboratory for Semi-Arid Climate Change of the Ministry of Education, College of Atmospheric Sciences, Lanzhou University, Lanzhou 730000, China
[10]Institute of Atmospheric Environment, China Meteorological Administration, Shenyang 110166, China
[11]School of Meteorology, University of Oklahoma, Norman, OK 73072, USA
[12]Beijing Municipal Institute of Labor Protection, Beijing 100054, China

**Correspondence:** Xuelei Zhang (zhangxuelei@neigae.ac.cn); Daniel Q. Tong (qtong@gmu.edu)

**Abstract:** Mineral dust particles play an important role in the Earth system, imposing a variety of effects on air quality, climate, human health, and economy. Accurate forecasts of dust events are highly desirable to provide early-warning and inform decision-making. East Asia is one of the largest dust sources in the world. This study applies and evaluates four widely used regional air quality models to simulate dust storms in Northeastern China. Three dust schemes in the Weather Research and Forecast with Chemistry (WRF-Chem) (version 3.9.1), two schemes in CHIMERE (version 2017r4) and CMAQ (version 5.2.1), and one scheme in CAMx (version 6.50), were applied to a dust event during May 4th~6th, 2015 in Northeastern China. Most of these models were able to capture this dust event, except CAMx which has no dust source map covering the study area, hence another dust source mask map was introduced to replace the default one for the subsequent simulation. Although these models reproduced the spatial pattern of the dust plume, there were large discrepancies between predicted and observed $PM_{10}$ concentrations in each model. In general, CHIMERE had relatively better performance among

all simulations with default configurations. After parameter tuning, WRF-Chem with the AFWA scheme using seasonal dust source map from Ginoux et al. (2012) showed the best performance, followed by WRF-Chem with UOC_Shao2004 scheme, CHIMERE, and CMAQ. The performance of CAMx had significantly improved by substituting the default dust map and removing the friction velocity limitation. This study suggested that the dust source maps should be carefully selected on regional scale or replaced with a new one constructed with local data. Moreover, further study and measurement on sandblasting efficiency of different soil types and locations should be conducted to improve the accuracy of estimated vertical dust flux in air quality models.

**Key words:** Air quality models, Dust, Forecast, Evaluation, Asia, Northeastern China

## 1. Introduction

Wind-blown dust is typically emitted from areas with dry, erodible surfaces, such as desert, Gobi and cropland, during high wind periods. It exerts significant effects on air quality (Giannadaki et al., 2014), atmospheric visibility (Mahowald et al., 2007), human health (Goudie, 2014; Zhang et al., 2016; Tong et al., 2017), ecosystem (Jickells et al., 2005; Schulz et al., 2012) and climate (Prospero and Lamb, 2003). Depending on the extent to which human activities are involved, dust emissions can be classified as natural or anthropogenic. Natural dust emissions are activated by wind from undisturbed surface in arid or semi-arid areas, such as the Sahara Desert in North Africa (Formenti et al., 2011), alluvial plains and deserts in West Asia and Central Asia (Cao et al., 2015; Xi and Sokolik, 2015), deserts and sandy lands in East Asia (Laurent et al., 2005), various desert landforms in Australia (Revel-Rolland et al., 2006) and deserts in the southwest USA (Gillette et al., 1996; Zhao et al., 2012). Anthropogenic dust emissions are either activated by mechanical forces (tilling, mining, etc), or by wind at surface disturbed by human activities. Agricultural activities that disturb the soil surface (such as tillage and reaping) can greatly increase the frequency and intensity of wind-blown dust (Zender et al., 2004; Guan et al., 2016). The erodible potential of farmlands depends strongly on agricultural management practices, such as timing of cropping and grazing, and soil conservation measures (Munkhtsetseg et al., 2017). A modeling study by Liora et al. (2016) showed that anthropogenic dust contributes approximately 10% of total $PM_{10}$ emissions in Europe. Using remote sensing observations, Ginoux et al. (2012) estimated that anthropogenic wind-blown dust sources account for 75% of emissions in Australia and 25% globally. Wind-blown dust emissions from cropland is of global importance (Mendez and Buschiazzo, 2010; Singh et al., 2012; Wang et al., 2013; Chappell et al., 2014; Xi and Sokolik, 2016).

Numerical dust models are often used to assess the magnitude of wind-blown dust emission and to predict its effects on air quality and climate. Several dust schemes to estimate the dust flux into the atmosphere and other relevant parameters have

been proposed in the past twenty years, and some have been coupled with air quality models, such as WRF-Chem (Kang et al., 2011; Su and Fung, 2015; Flaounas et al., 2017) , CMAQ (Wang et al., 2012; Foroutan et al., 2017), CAMx (Klingmuller et al., 2018), CHIMERE (Menut et al., 2013; Mailler et al., 2017), ALADIN-SURFEX (Mokhtari et al., 2012), LOTOS-EUROS (Manders-Groot et al., 2016), EMEP MSC-W (Simpson et al., 2012), NAQPMS (Li et al., 2012) and CUACE/Haze (Wang et al., 2015). These models are widely used to study the air quality and climate effects of dust emissions. Most model applications, however, only adopt dust schemes designed for natural wind-blown dust from arid areas. It is unclear how well these models perform in areas with active agricultural operations.

The selection and usage of dust emission schemes and their input datasets are very important in establishing reliable air quality prediction. The evaluation and validation of dust emission schemes and relevant datasets in different air quality models on a continental scale has been carried out for East Asia (Dong et al., 2016), West Asia (Nabavi et al., 2017; LeGrand et al., 2019), North America (Foroutan et al., 2017), Europe, Northern Africa and the Middle East (Menut et al., 2013; Flaounas et al., 2017; Rizza et al., 2017). Nevertheless, evaluation of the regional performance of different air quality models and dust schemes remains inadequate.

Many previous multi-model evaluation studies focused on the climatic implications of different dust schemes at both global and regional scales. A comprehensive evaluation of 14 global aerosol models reported that the estimated dust emissions in Asia vary widely, ranging from 27 to 873 Tg/year (Huneeus et al., 2011). Two different dust emission schemes in EMAC (ECHAM5/MESSy2.41 Atmospheric Chemistry model) were shown to produce similar atmospheric dust loads in North Africa, but differ considerably in Asia, the Middle East and South America (Astitha et al., 2012). Ridley et al. (2016) reported that the global simulated aerosol optical depth (AOD) may vary by over a factor of 5 among four global models, and that dust emissions in Africa are often overestimated at the expense of emissions from Asia and the Middle East; in addition, dust was removed too rapidly in most models. On the regional scale, Todd et al. (2008) showed that the simulated dust flux and concentration from five models differed by at least one order of magnitude during a 3-day dust event over the Bodélé depression in northern Africa. Evan et al. (2015) demonstrated a power law relationship between modeled dust emission frequency and dust emission intensity in four regional models for North Africa. Huneeus et al. (2016) evaluated five dust forecast models during an intense Saharan dust outbreak affecting western and northern Europe in April 2011, noting that all models were better at predicting AOD than near-surface dust concentration over the Iberian Peninsula and tended to underestimate the long-range transport of dust. An evaluation of eight regional dust models with various dust emission schemes and other configurations was conducted for East Asia by Uno et al. (2006). Their results demonstrated that the models could correctly capture the major dust onset and cessation timing, but the maximum concentration of each model differed by a factor of 2-4. Several other studies have focused on the assessment of one or more dust schemes in the

WRF-Chem model, over regions such as the Mediterranean (Flaounas et al., 2017), Middle East (Prakash et al., 2015), and Central and East Asia (Darmenova et al. 2009; Xi and Sokolik, 2015). Dust modeling requires sufficient parametrization, high-quality input data and practical tuning techniques to enable results to best match observations (Basart et al., 2012; Flourous, 2017).

Accurate forecasts of dust emissions and transport are demanded globally by society, to address many health and economic issues, especially air quality. Here we present a comprehensive evaluation of multi-model simulations of windblown dust emissions in air quality models during a dust episode in East Asia, using a number of dust emission schemes with four state-of-the-art air quality models. East Asia is one of the world's largest dust sources, contributing about 30% of total global dust loading. This study focuses on Northeastern China, a unique dust source region with varying land use types,

including deserts, semiarid land, and croplands. In addition, this region is known for diverse soil texture and organic content. The Northeast China Plain, the nation's breadbasket, is made of soil with abundant organic matter. Dust storms originated in this region are often called "Black sandstorm" (Zhang et al., 2015). There are also areas with saline-alkali soil on the western side of this region, giving dust storms the white color ("white sandstorms"). This region is also known to experience high wind during springtime. All of these characteristics present challenges for numeric models to predict dust storms, making

Northeastern China an ideal region for assessing the capability of dust models. We choose four air quality models (CMAQ v5.2.1, WRF-Chem v3.9.1, CHIMERE v2017r4 and CAMx v6.50), each configured with a selection of dust emission schemes and source maps to simulate a well-observed regional pollution event with strong dust influence. Detailed description of the study region, model configuration and dust schemes are presented in Section 2. Comparisons of dust schemes and dust source maps are described in Section 3. Results of model simulations and verification with the

ground-based and satellite-based observations are presented and discussed in Section 4. We conclude in Section 5.

## 2.    Model configuration, observations and methods

### 2.1   Study area and model domain

Northeastern China (NEC) (38°42′–53°33′ N and 115°31′–135°2′ E) is located at the eastern end of the northern hemisphere dust belt. This area covers about 1.47 million square kilometers, accounting for about 15% of the Chinese land

area (Fig. 1). NEC has a semi humid continental climate with prevailing westerly winds throughout the year. A major grain production region in China, NEC includes the alluvial Northeast China Plain with farmlands characterized by mollisol (Udolls, USDA Soil Taxonomy, or Black Chernozem, Canadian soil classification). Due to the long cold season and strong spring winds, the exposed cropland is vulnerable to wind erosion (Dickerson et al., 2007). Two of the four major Chinese

sandy lands, the Horqin and Hulun Buir sandy lands, are located in the western NEC while several other sandy/barren land regions are located in the central area and surrounded by cropland. The Gobi Desert between China and Mongolia is located to the west of NEC.

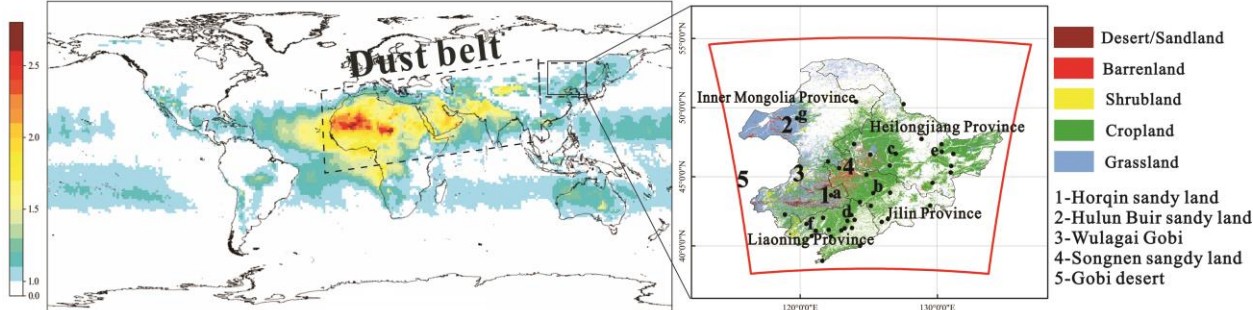

**Figure 1.** The global aerosol index distribution and the dust belt location (dashed rectangle) as described in Varga (2012), and the geographical coverage of the NEC domain on the right. Dots in the NEC domain represent monitoring sites (Labels a~g indicates the monitoring sites at Tongliao, Changchun, Harbin, Shenyang, Jiamusi, Jinzhou and Hulun Buir.)

The model domain centers on 46.715 °N, 125.081 °E and is defined on a lambert conformal projection. The true latitudes of the domain are 30 °N and 60 °N, and composed of $60 \times 73$ grid cells with a horizontal grid resolution of 25 km $\times$ 25 km and 30 vertical levels. The domain covers the whole of NEC (as shown by the dark blue line in Fig. 1). The initial and boundary fields were obtained from the final (FNL) operational global analysis data of the National Center for Environmental Prediction (NCEP) with a horizontal resolution of 1 °$\times$ 1 °, updated every 6 h (http://rda.ucar.edu/datasets/ds083.2).

### 2.2 Observational data sources

Air quality monitoring data were acquired from the national air quality history database (http://beijingair.sinaapp.com), which contains hourly data and information from China's national environmental monitoring center. The hourly monitoring data used in this study include $PM_{10}$ and $PM_{2.5}$ concentrations in 40 cities of NEC (Fig. 1) for the time period from May 3[rd] to May 7[th], 2015. Deep Blue Aerosol Optical Depth (AOD) data were obtained from the MODIS-Aqua with a resolution of 10 km$\times$10 km from the archive of NASA Level-1 and Atmosphere Archive & Distribution System (LAADS) (https://ladsweb.modaps.eosdis.nasa.gov/archive). In addition, data from the Cloud-Aerosol Lidar and Infrared Pathfinder Satellite Observation (CALIPSO) satellite were used to investigate the vertical distribution of transported dust particulates.

### 2.3 The Springtime Dust Episode

The dust event on May 5, 2015 was selected for model evaluation in this study from examining the time series of observed

$PM_{10}$ concentrations. Satellite images indicate that large areas of central NEC were covered by higher AOD during the May 5[th] dust event when compared to the preceding two days (Fig. 2a~c). The mean AOD in central NEC quickly increased from 0.6 on May 4[th] to >1.0 on May 5[th], while AOD in other regions was relative lower (<0.3). This indicated that the event was not caused by long-distance transported dust from western China and Mongolia, but instead was a locally generated event. Meanwhile, the vertical distributions of aerosol subtypes derived from CALIPSO observations indicated that dust was distributed in the atmosphere below 1 km in NEC, acted as the primary pollutant on May 5[th].

PM concentrations observed from 40 air quality monitoring sites over NEC were used to analyze the spatiotemporal distribution of the dust plumes. The spatial distribution of daily $PM_{10}$ concentrations during May 5[th], 2015 was consistent with the retrieved AOD (Fig. 2d). This event originated in the region around Tongliao (such as the Horqin sandy land and saline-alkali soil) on 20:00 UTC of May 4[th], and lasted for nearly 17 hours.

To facilitate comparison and evaluation of these air quality models, we divided the study region into four areas according to $PM_{10}$ levels: heavy dust central area (CTA), northwest moderate dust area (NWA), and two light dust areas in the northeast (NEA) and southwest (SWA) (Fig. 2e). Daily concentrations of $PM_{10}$ at the central sites, such as Tongliao and Changchun, exceeded 700 $\mu g\ m^{-3}$. The concentrations of $PM_{10}$ in NE and SW ranged from 100 to 500 $\mu g\ m^{-3}$. All these values considerably exceeded the $PM_{10}$ level-2 concentration limits (150 $\mu g\ m^{-3}$) of the NAAQS (National Ambient Air Quality Standard). On May 5[th], the ratio of $PM_{2.5}/PM_{10}$ was 0.14, indicating that the particulate matter was dominated by coarse dust particles (Tong et al., 2012), consistent with the aerosol subtype observations of CALIPSO (Fig. 2f).

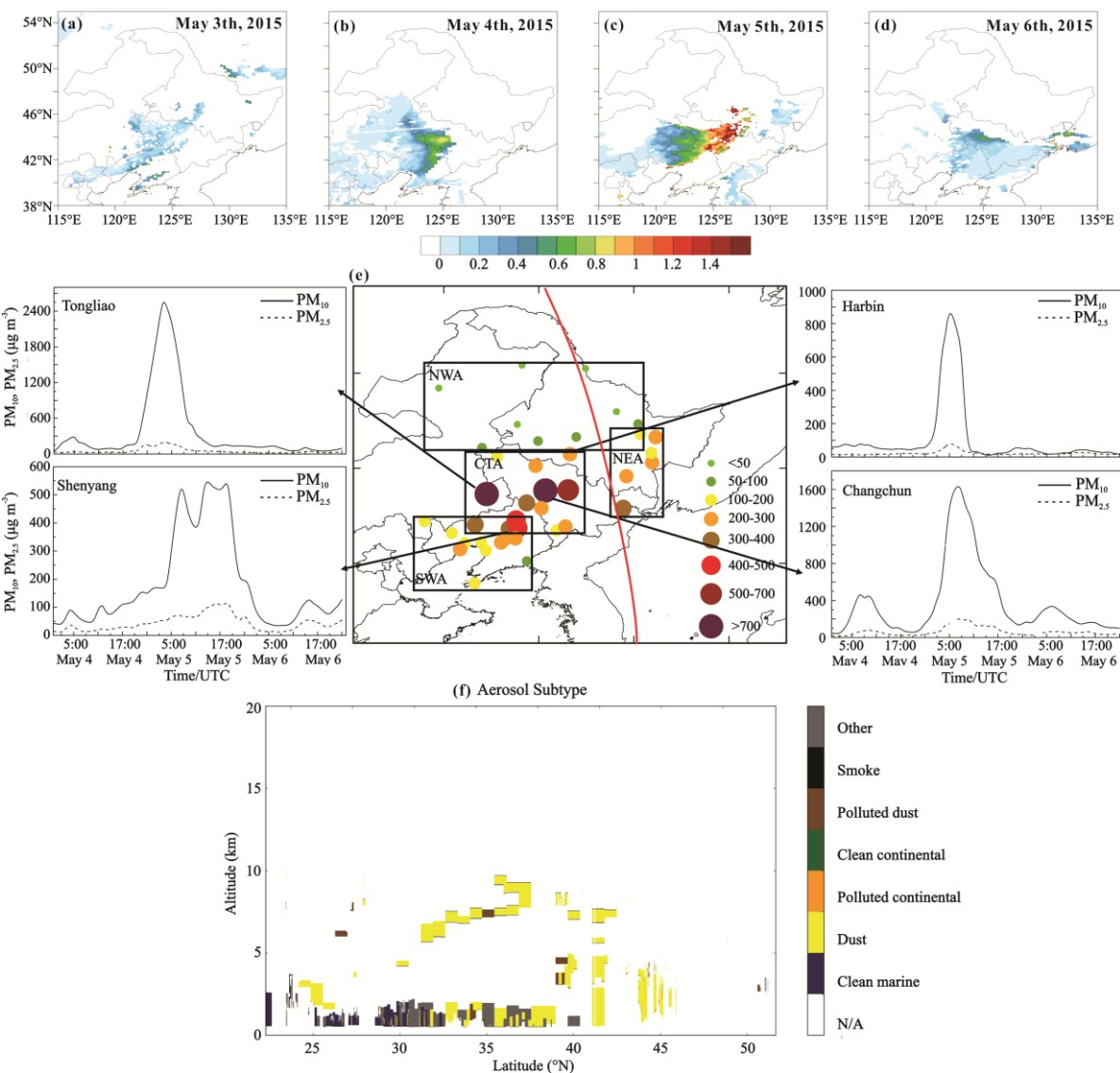

**Figure 2.** Satellite and ground observations of the May 5, 2015 dust event over Northeastern China: (**a**)–(**d**) Daily MODIS aerosol optical depth (AOD) at 550 nm before, during and after the storm; (**e**) daily mean $PM_{10}$ concentrations (µg m$^{-3}$) measured at four ground sites (Tongliao, Changchun, Harbin and Shenyang) on May 5th, 2015 and hourly $PM_{10}$ (solid line) and $PM_{2.5}$ (dashed line) variations in during the dust period. The blue line in (**e**) indicates the path of the CALIPSO satellite, and CALIPSO aerosol subtype between 4:30 and 4:43 UST (**f**).

## 2.4 Description of air quality models and dust schemes

This study focuses on comparing dust emission schemes in four air quality models as described below.

### 2.4.1 Dust schemes in WRF-Chem v3.9.1

The Weather Research and Forecast community model coupled with a chemistry model (WRF-Chem) is a coupled online community model able to simulate gas and aerosol chemistry simultaneously with the meteorological fields, and is generally used for the prediction and simulation of weather, air quality and regional climate from cloud scales to regional scales (Grell et al., 2005). The WRF-Chem version 3.9.1 was used in this study. Three dust schemes, GOCART, AFWA and UOC, are tested here. The latter scheme was further divided into three dust emission parameterizations with various levels of complexity, namely Shao2001, Shao2004 and Shao2011 (Shao, 2001; Shao, 2004; Shao et al., 2011).

In the GOCART scheme, the dust emission is based on an equivalent empirical formulation by Gillette and Passi (1988) which requires data on the wind speed at 10 m and a threshold velocity to initiate wind erosion, as well as the surface erodibility (Ginoux et al., 2001). Comparing to other dust emission schemes, the dust emission flux in this scheme can be simply and directly calculated via the variables like wind speed, soil moisture and air density (which can be obtained from most numerical weather models) over source emission areas within the dust source map, without the conversion from horizontal to vertical flux (Fig. S1). The AFWA dust scheme is a modified version of Marticorena and Bergametti (1995) dust scheme developed by the Air Force Weather Agency (LeGrand et al., 2019). Unlike GOCART scheme, friction velocity is introduced into this scheme and the dust emission is calculated as a saltation flux and vertical uplift dust flux, which is proportional to the horizontal saltation flux, based on the soil clay content (Marticorena and Bergametti, 1995). A soil moisture correction term is also applied to the threshold friction velocity, however, this term in AFWA scheme is calculated according to the method described by Fécan et al. (1999) (Fig. S2), which is different from that used in the GOCART scheme. The UOC (University of Cologne) scheme accounted for the saltation bombardment, aggregate disintegration and volume removal of saltating particles. The vertical dust emission flux is proportional to horizontal saltation flux, but the ratio significantly depends on soil texture and soil plastic pressure (Shao, 2004). The fully disturbed soil particle size distribution was omitted in the simplified scheme of Shao2011 (Shao et al., 2011). The parameterization of Shao2004 has been verified by field observations and was therefore adopted in this study. Unlike the GOCART and AFWA dust emission schemes, the threshold friction velocity is obtained via the method from Shao and Lu (2000) rather than Bagnold (1941). Although the equation of moisture correction in UOC scheme is also from Fécan et al. (1999), it is based on the volumetric soil moisture and empirical constants as a function of soil texture (Klose et al., 2014). Furthermore, an additional correction term, roughness correction (or drag partition correction), is also introduced to describe the influence of non-erodible elements (such as vegetation, peddle etc.) on the threshold friction velocity (Raupach, 1992) (Fig. S3). In addition, the UOC scheme only uses the erodible area to constrain the dust source locations instead of scaling dust emissions. Note that the last term in

the saltation flux formula in UOC source code is mistakenly expressed as $(1 + (\frac{u_{*t}}{u_*})^2)$ in WRF-Chem before the version of

4.0. In this study, it has been corrected into $(1 + \frac{u_{*t}}{u_*})^2$ in WRF-Chem version 3.9.1 according to the description in Shao et

al. (2011). This revision could increase the saltation flux by a factor of 2 or more. More detailed physical descriptions and

defects in source codes of above three dust schemes in WRF-Chem model had been explicitly documented, and all schemes

had been evaluated over southwest Asia in LeGrand et al. (2019).

### 2.4.2 Dust schemes in CHIMERE v2017r4

CHIMERE is an Eulerian off-line chemistry-transport model covering local to continental scales (from 1 km to 1 degree resolution). An aerosol module was implemented into CHIMERE in 2004 with further modifications concerning the natural dust emissions and resuspension over the northern Atlantic and Europe (Vautard et al., 2005; Hodzic et al., 2006). Dust emissions have been verified for long-distance transported dust by comparison with long-term and field measurements (Schmechtig et al., 2011; Bessagnet et al., 2017). The CHIMERE version 2017r4 was used in this study.

Three dust emission schemes were employed in the CHIMERE model: the MBW scheme (White, 1986; Marticorena and Bergametti, 1995), AGO scheme (Alfaro and Gomes, 2001; Menut et al., 2005) and KOK scheme (Kok et al., 2014a). Extension of the dust production model to any domain over the globe was available since the model version of chimere2016a, and the KOK scheme was also implemented in this version. In the MBW scheme, the vertically integrated saltation flux was estimated using the equation introduced by White (1986). The vertical dust flux in the second scheme was computed based on the partitioning of the kinetic energy of individual saltating aggregates and the cohesion energy of the populations of dust particles with the assumption that dust emitted by sandblasting is characterized by three modes whose proportion depends on the wind friction velocity. The vertical dust flux in the KOK scheme was estimated directly without converting from horizontal flux to vertical flux but only controlled by dust emission coefficients, namely, bare soil fraction, soil clay fraction, surface friction velocity and threshold friction velocity (Kok et al., 2014a). The dust schemes in CHIMERE follow similar calculating process as the UOC scheme, and the flow chart for these schemes is showed in Fig. S4. Moreover, there are two options for calculation of threshold friction velocity, Iversen and White (1982) and Shao and Lu (2000) in CHIMERE, and it uses the equation from Marticorena et al. (1997) to calculate the roughness correction. Additionally, it needs to note that the friction velocity is calculated independently in this model and equation of Weibull distribution is applied for wind speed adjustment (Cakmur et al., 2004; Pryor et al., 2005). According to the CHIMERE source code, all three schemes needed external land-surface static data (such as land use type, soil type/fraction and vegetation cover) for the erodibility factor

calculation.

### 2.4.3 Dust schemes in CMAQ v5.2.1

The Community Multiscale Air Quality (CMAQ) model is a 3-D Eulerian photochemical dispersion model that allows for an integrated assessment of gaseous and particulate air pollution over many scales ranging from sub-urban to continental (Byun and Schere, 2006). CMAQ version 5.2.1 was used in this study.

The first wind-blown dust emission scheme, named FENGSHA, was implemented into CMAQ version 5.0 in 2012. Four land use types (barren land, shrub-grass land, shrub land and cropland) were treated as potential erodible dust sources instead of dust source maps, and the dust vertical flux was calculated according to a modified Owen's equation (Owen, 1964) when the friction velocity ($u_*$) exceeded the threshold friction velocity ($u_{*t}$) (which was set as constant value for each potential erodible land use type and soil texture based on literature and field measurements, such as 0.63 m s$^{-1}$ for clay loam of barren land). The effect of agricultural activities was calculated via a crop calendar, allowing the vegetation fraction to vary with the change of date. In the calculated dust emission from croplands in CMAQ, the vertical flux was improved by adding another two factors: the crusting factor $f_{cs}$ and tillage-ridge factor $f_{tr}$ (Zhang et al., 2015).

Another dust emission scheme was applied in CMAQ version 5.2 in 2017. In this scheme, the vertical dust emission flux was acquired based on the calculated horizontal dust flux (White, 1979) and sandblasting efficiency. The threshold friction velocity was calculated following Shao and Lu (2000), and the friction velocity was calculated based on an updated dynamic relation for the surface roughness length relevant to small-scale dust generation processes (Foroutan et al., 2017) (Fig. S5). Besides the potentially erodible land use types which were the same as those in the original FENGSHA module, the satellite-observed fraction of absorbed photosynthetically active radiation (FPAR) was introduced to act as a surrogate for vegetation cover fraction to constrain dust emission.

### 2.4.4 Dust scheme in CAMx v6.50

The Comprehensive Air quality Model with extensions (CAMx) is an Eulerian chemistry-transport model that allows for an integrated "one-atmosphere" assessment of gaseous and particulate air pollution ranging from urban to continental scales. The inline dust emission module had not been implemented into the newly-released CAMx version 6.50, but it had been fully developed as a pre-processing program (namely wbdust) which was used to provide binary dust emission files and to merge them with the emissions from other sources into model-ready emission files. The dust emission scheme in CAMx was based on a revised mineral dust emission scheme in the atmospheric chemistry–climate model EMAC (Astitha et al., 2012; Klingmuller et al., 2018). We obtained the source code of this dust scheme through private communication with Dr. Yarwood

Greg. Its vertical dust emission flux was calculated via the saltation flux and sandblasting efficiency when friction velocity exceeds a threshold value (Marticorena and Bergametti, 1995), similar to the MBW scheme in CHIMERE. The major improvements and adjustments were omitting the term supposed to account for the effect of soil moisture on dust emission, adding a topography factor which accounted for enhanced emissions from basins and valleys, filtering the sandblasting

efficiency of the soil clay fraction by a Gaussian function with an interquartile range of 5%, and limiting the maximum value of the friction velocity to 0.4 m s$^{-1}$ (Klingmuller et al., 2018). The schematic diagram of dust emission module in CAMx, as well as those in WRF-Chem, CHIMERE and CMAQ, are all provided in Fig. S1~S6 of the supplementary file.

## 2.5   Model configuration

### 2.5.1   Physical parameterization

The Weather Research and Forecasting (WRF) model version 3.9.1 was used to conduct the meteorological simulations, then to provide the hourly meteorological output fields to drive the air quality models of CHIMERE, CMAQ and CAMx while the chemistry module of WRF (WRF-Chem) was conducted simultaneously with the meteorological fields. As the surface wind speed was the dominant factor controlling dust blowing and transportation, its accuracy could significantly influence the results of dust modeling. Furthermore, the land surface characteristics played an important role in the WRF

surface wind simulation. For the purpose of comparing and selecting the optimal scheme (Table 1) to be used in the following dust emission simulations in the air quality models, two scenarios with different land-surface schemes (Noah-MP scheme and Pleim-Xiu scheme) were chosen for comparison. More detailed comparisons are provided in Section 2 of the Supplementary Information, and Scenario 2 was finally selected for the WRF model.

The FNL and static geographical fields are interpolated to the model domain resolution of 25 km $\times$ 25 km by using the WRF

preprocessing system (WPS). In addition, the land use and soil category datasets used in this study were obtained from WPS static data file website (http://www2.mmm.ucar.edu/wrf/src/wps_files/), IGBP-Modified MODIS land use data was selected for the simulations of WRF-Chem, CMAQ and CAMx model while USGS dataset was used for CHIMERE as the dust model in CHIMERE could only read USGS data for dust emission calculations.

**Table 1.** WRF parameterization settings

| Physical scheme | Scenario 1 | Scenario 2 |
|---|---|---|
| Microphysics | WRF double moment, 6-class scheme | |
| Longwave radiation | rrtmg scheme | |
| Shortwave radiation | rrtmg scheme | |

| Surface layer | Revised MM5 Monin-Obukhov | Pleim-Xiu scheme |
|---|---|---|
| Land-surface | Noah-MP land-surface model | Pleim-Xiu scheme |
| Number of soil layers | 4 | 2 |
| Boundary layer | YSU scheme | ACM2 (Pleim) scheme |
| Cumulus parameterization | Grell-Devenyi ensemble scheme | |

### 2.5.2 Chemical parameterization

WRF-Chem v3.9.1 simulations were executed with different source maps (G01, G01_1.0, K08, G12 and MDB) for each dust scheme, GOCART, AFWA, and UOC_Shao2004. The chemistry scheme with chem_opt of the GOCART simple aerosol scheme was used without anthropogenic emission input. CHIMERE v2017r4 was used with the MELCHIOR chemistry mechanism and MBW, AGO and KOK dust emission schemes along with 3 algorithms of erodible fraction, as well as no surface anthropogenic emissions. The Iversen and White (1982) and Shao and Lu (2000) methods (IW and SL hereafter) were chosen as algorithms for calculation of the threshold friction velocity.

As for CMAQ v5.2.1, the CB6R3 gas-phase mechanism and AE6 aerosol mechanism with sea salt and speciated PM aqueous/cloud chemistry (cb6r3_ae6_aq) were used in this study. The inline dust emission calculation was executed with and without agricultural activity (CTM_ERODE_AGLAND).

The dust emission module used the meteorological output fields to obtain a gridded dust emission flux, and then the emitted dust flux was reformatted and merged with anthropogenic emissions for CAMx. CAMx v6.50 with the CB6r2 gas-phase mechanism and AE6 aerosol mechanism was used in this study.

All simulations covered the 96 hours from 0:00 May 3[th] to 23:00 May 6[th], 2015 and the first 24 hours was regarded as model spin-up. And more detailed configuration information and the namelist files for model simulations were provided on https://doi.org/10.5281/zenodo.3376774.

### 3. Results and discussion

### 3.1 Comparison of dust source maps and erodibility fractions

### 3.1.1 Comparison of dust source maps in WRF-Chem v3.9.1 and CAMx v6.50

The default dust source map (or dust source function) with a horizontal resolution of 0.25 °×0.25 ° used in WRF-Chem v3.9.1, was obtained from static geographical datasets (Ginoux et al., 2001). The map comprises the gridded fraction of alluvium available for wind erosion, calculated from topography and elevation. In this study, we named this source map as

G01_0.25 according to first author and published year of relevant literature and its spatial resolution. It   shows only one

weakly erodible area in NEC, located in the Horqin sandy land, with erodibility fraction values < 0.2 (Fig. 3a). Because the

source map plays a critical role in determining the spatial distribution of dust emission and the calculated magnitude of dust

fluxes, here we test five other dust source maps to test with these models (see Table 2 for more details). The source map with

resolution of 1 °×1 ° developed by Ginoux et al. (2001) (namely G01_1.0) basing on the same method with G01_0.25 was

obtained from the homepage of Dr. Paul Ginoux (https://www.gfdl.noaa.gov/pag-homepage). Except G01_0.25 and G01_1.0,

the other source maps were obtained using satellite observations. The map with resolution of 0.25 °×0.25 °provided by Koven

and Fung (2008) (K08_0.25 hereafter) was calculated via the relationships between landscape characteristics, residual

landscape roughness and aerosol optical depth, and presented as globally-available erodible fractions. A global-scale high

resolution (0.1 °) dust source product was derived from a combination of a climatological analysis of MODIS Deep Blue

AOD data and land use data (Ginoux et al., 2012). Sources were classified as natural or anthropogenic (primarily

agricultural) and their global distributions were described by frequency-of-occurrence (FO). In this study, we established a

simple conversion from FO value into erodible fraction: FO(0.05)→0.15, FO(0.1)→0.3, FO(0.2)→0.4, FO(0.25)→0.5,

FO(0.4)→0.7, FO(0.5)→0.8, FO(0.6)→0.9. The source map with only natural origins was named G12_0.1_natural while

that with both natural and anthropogenic origins was named G12_0.1_ant+nat.

The source map of G01_1.0 evidently had more widespread erodible lands than that of G01_0.25, regardless of region or

erodible fraction values (Fig. 3b). Four erodible areas were depicted in the dust source map of K08_0.25: the central plain

area of NEC, Hulun Buir sandy land, the northeast corner of NEC and the North China Plain (Fig. 3c). It was obvious that

the croplands were identified as dust source areas in K08_0.25, although the erodibility was likely overestimated (e.g., it can

not be higher than that of the sandy lands as shown in this map). This is due to the difficulty to distinguish anthropogenic

emitted particulates, such as industrial emissions, from the AOD used for retrieving dust sources. Moreover, the erodible

fraction values in the northeast corner were significantly overestimated, for example those in Xingkai (Khanka) Lake

(45.33 °N, 132.67 °E) and its surroundings did not have any erodible potential according to our ground survey. The spatial

distribution of dust sources in G12_0.1_natural in this area seems to miss many dust source areas, such as the Horqin sandy

land and Hulun Buir sandy land (Fig. 3d). The dust source map obtained from the NASA-Unified WRF (NU-WRF) version

7 (Kim et al., 2014) was similar to the former source map but divided into four seasons, therefore it was named as

G12_0.1_seasonal in this study (Fig. 3f). The spring spatial distribution of dust sources was used in this study. In comparison,

G12_0.1_ant+nat and G12_0.1_seasonal had a similar spatial pattern to that of K08_0.25 but with lower fraction values (Fig.

3c, 3e and 3f), as they were retrieved from the same satellite products. The pattern of the spatial distribution was reasonably

30   similar to the erodible land use type distribution shown in Fig. 1.

**Table 2.** Information on dust source maps used in WRF-Chem

| Name | Method | Region | Resolution | Time | References |
|------|--------|--------|------------|------|------------|
| G01_0.25 | Topographic depression | Global | 0.25 ° | Constant | Ginoux et al (2001) |
| G01_1.0 | Topographic depression | Global | 1 ° | Constant | Ginoux et al (2001) |
| K08_0.25 | Satellite AOD, levelness and residual landscape roughness | Global | 0.25 ° | Constant | Koven and Fung (2008) |
| G12_0.1_natural | Satellite AOD and frequency of dust occurrence | Global | 0.1 ° | Constant | Ginoux et al. (2012) |
| G12_0.1_ant+nat | Satellite AOD | Global | 0.1 ° | Constant | Ginoux et al. (2012) |
| G12_0.1_seasonal | Satellite AOD | Global | 0.1 ° | Seasonal | Ginoux et al. (2012) |

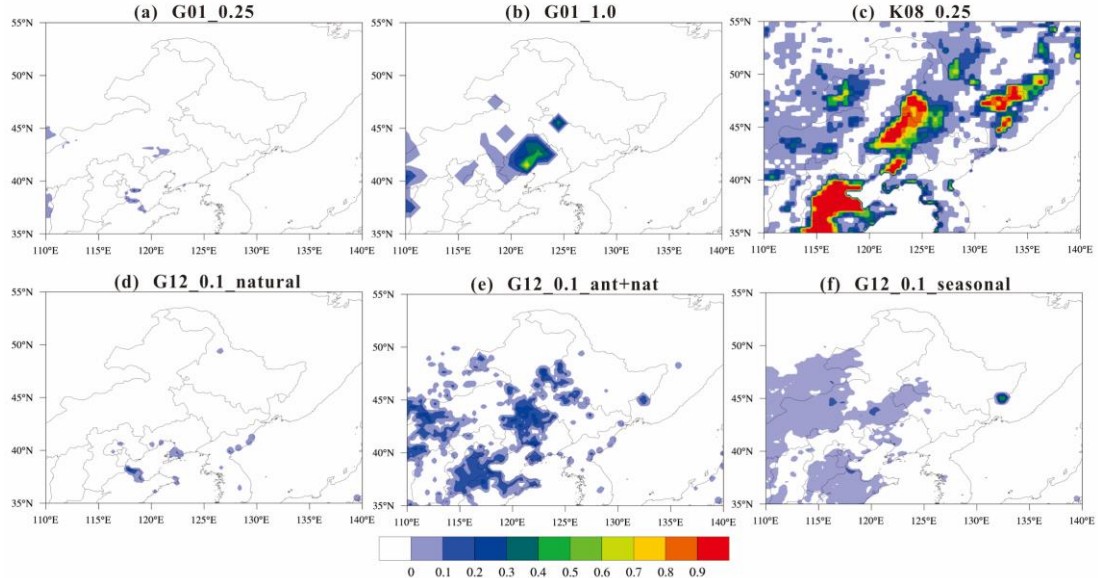

**Figure 3.** Dust source maps in NEC. (**a**) G01_0.25, (**b**) G01_1.0, (c) K08_0.25, (**d**) G12_0.1_natural, (**e**) G12_0.1_ant_nat, (**f**) G12_0.1_seasonal (spring))

5    A dust mask file was used in CAMx v6.50, which only had two values: 0 indicating no erodible dust potential while 1 dust emitting capacity in the grid cell. Dust flux was then calculated with the clay fraction-dependent vertical-to-horizontal dust flux ratio (Fig. S8a). Unfortunately, no dust erodible area was recorded for the NEC region in the dust mask file (Fig. S8b). Therefore, a dust source map will be introduced and used instead of the original dust mask file for further evaluations of the dust emission scheme in CAMx.

### 3.1.2 Comparison of erodible land fractions in CMAQ v5.2.1 and CHIMERE v2017r4

Instead of using a prescribed dust source map, the dust schemes in CMAQ v5.2.1 and CHIMERE v2017r4 estimated erodible land fraction based on land use, crop types, and/or crop calendar. Erodible land fraction in CMAQ was calculated by multiplying the fraction of erodible land use type with an erodibility potential factor assigned to each land use type. In this method, four land use categories (shrub land, shrub grass, cropland and sparse barren land) were considered as erodible land types when the Biogenic Emissions Landcover Database version 3 (BELD3) dataset was used during regional simulation in the USA. Otherwise, the global MODIS FPAR data were recommended to represent the vegetation fraction (Flaounas et al., 2017). However, the grassland fraction was not taken into account when using the USGS or MODIS land use dataset according to the source code of CMAQ, which may lead to an underestimation of dust emission. Thus, this kind of land use type was added into the source code file of LUS_DEFN.F using the same erodible fraction as that of a similar land use type (shrub-grass) in BELD3 (with an erodibility potential value of 0.25), and the final, modified distribution of erodible fraction is depicted in Fig. 4a.

Three methods were used for calculating erodible land fraction in the CHIMERE model. When erodibility option *ierod* was set to 1 in the model, the USGS land use data were used, and the erodible fraction depended on the fractional area without vegetation if the land use type was cropland. The fraction was set to 1 for shrub and barren lands. The second method (*ierod* = 2) simply depended on an erodibility value retrieved from monthly erodibility data derived from MODIS surface reflectance stored in the CHIMERE static dataset. The last method was the mixed usage of USGS and MODIS (*ierod* = 3): when land use type was cropland, the fraction was calculated following method 1, otherwise it was set to the MODIS erodibility for shrub and barren lands. Note that *ierod* = 3 was the default option for dust emission in CHIMERE. Figure. 4(b~d) shows that the erodible fractions acquired from the three methods had similar distributions, although values in method 2 were considerably lower than those of the other two. In addition, the user guide for CHIMERE noted that the USGS land use type must be used to ensure the erodible land fraction is calculated correctly.

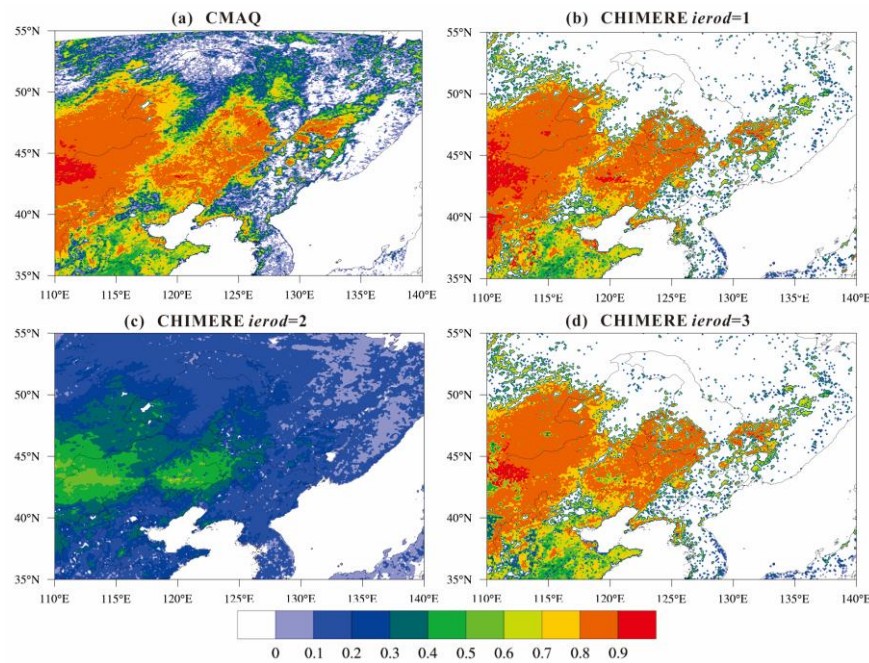

**Figure 4.** Maps of erodible fractions in NEC in CMAQ (**a**) and CHIMERE model (**b**) uses USGS land use (*ierod*=1), (**c**) uses MODIS surface reflectance (*ierod*=2), (**d**) is a mix of USGS and MODIS (*ierod*=3)

## 3.2 Performance of WRF-Chem v3.9.1 dust simulation

WRF-Chem v3.9.1 simulations showed large differences among different dust schemes and source maps in terms of both spatial distributions and values of dust emissions. $PM_{10}$ simulated by GOCART scheme with default source map (G01_0.25) presented relatively low concentrations with daily averages less than 40 μg m$^{-3}$, mainly concentrated in the Horqin sandy land in Tongliao and extending to the east of Liaoning Province with concentrations less than 20 μg m$^{-3}$ (Fig. 5a). In comparison, the spatial distribution of daily $PM_{10}$ concentration simulated by GOCART with the G01_1.0 source showed three dominant dust emission areas: Horqin; the border between western Jilin and Heilongjiang Province; and coastal Liaodong Bay (Fig. 5b). The higher simulated $PM_{10}$ concentration in the latter area was not supported by observations. The high concentration centers around Tongliao and in eastern Liaoning Province reached values of 600 μg m$^{-3}$, but dust was transported directly to the west without reaching the cities of Changchun and Harbin. The distribution of $PM_{10}$ using the GOCART scheme with the K08_0.25 source map yielded concentrations in Horqin sandy land and Songnen sandy land and their surrounding areas. The simulated concentrations with K08_0.25 were greater than those simulated by the GOCART scheme with other source maps (Fig. 5a~f), but two-fold of the observed values. The result of G12_0.1_natural (Fig. 5d)

showed dust emissions in coastal Liaoning Province with quite low dust intensity, showing that this natural source distribution was not applicable in this area. The $PM_{10}$ patterns with G12_0.1_ant+nat and G12_0.1_seasonal indicated that their dust source regions were similar (Fig. 6e and 6f), and the simulated daily concentration of G12_0.1_seasonal was only about 100~200 μg m$^{-3}$, compared to the observed concentrations of 100~700 μg m$^{-3}$.

Similar spatial distributions of $PM_{10}$ were obtained using the AFWA and UOC_Shao2004 schemes with the above 6 NEC dust source maps, but the simulated $PM_{10}$ concentrations using each source map was more than 1 order of magnitude greater than those of GOCART (Fig. 5g~5r). Furthermore, the spatial patterns of dust simulated by UOC_Shao2004 with the last two source maps (Fig. 5q~5r) extended further northeastwards than those with GOCART and AFWA, and were more consistent with the observations. The overestimation of AFWA scheme might in part be explained by the fact that the AFWA scheme considered vertical dust flux only related to the clay content, unlike the UOC scheme which considered it to be inversely proportional to surface hardness (Kang et al., 2010; Rizza et al., 2016; Rizza et al., 2017). Meanwhile, the misuse in number and distribution of saltation size bins in AFWA might also be part of the reason. The last three bins of the total nine saltation size bins were sand-sized bins and they were also configured to constitutes all of the possible sand mass fractions which indicated that the sand in the soil surface was entirely composed of fine sands, resulting in the increase of the strength of the saltation bin-specific weighting factors and emission of the dust particles (LeGrand et al., 2019). In addition, we found that the simulated dust concentration with these three schemes generally presented over-predictions in this area, this might because of the usage of Pleim-Xiu (PX) land surface scheme. An additional dust simulation with Noah land surface scheme was conducted and the results showed lower $PM_{10}$ concentration (Fig. S9), and it also indicated similar wind speed and higher surface soil moisture (Fig. S10, Table S7) than the simulated values using PX scheme. The higher soil moisture resulted in increasing the value of soil moisture correction, which was used for calculating threshold friction velocity, by about 10%. These discrepancies may result in the differences of estimated dust emissions and it could be the reason of the stronger dust emission when using the PX scheme.

As mentioned above, we found that the distribution and intensity of modeled dust aerosols were sensitive to the dust source maps in use. Further analysis of Figs. 3 and 5 shows that the source maps of G01_0.25 and G12_0.1_natural were not able to reproduce this dust event in NEC. Dust source regions in other source maps were more or less similar, generally located in Horqin sandy land, mid-west Jilin Province and coastal Liaodong Bay. Observations suggest the modeled dust source in Liaodong Bay might be inaccurate, as measured concentrations were relatively low in that area.

Since results obtained by all three dust emission schemes with four source maps (G01_1.0, K08_0.25, G12_0.1_ant+nat, and G12_0.1_seasonal) were in better agreement with observations, these maps were used in subsequent evaluation. Note that the exceedingly high $PM_{10}$ calculated via AFWA and UOC_Shao2004 or with the K08_0.25 source map indicated that a

tuning coefficient was needed to improve the model performance. Considering that the source maps of G12_0.1_ant+nat and G12_0.1_seasonal were obtained via the same methodology but the latter one provides seasonal divisions making it more reasonable and closer to the actual environment, G12_0.1_seasonal was chosen for the next step in the evaluation.

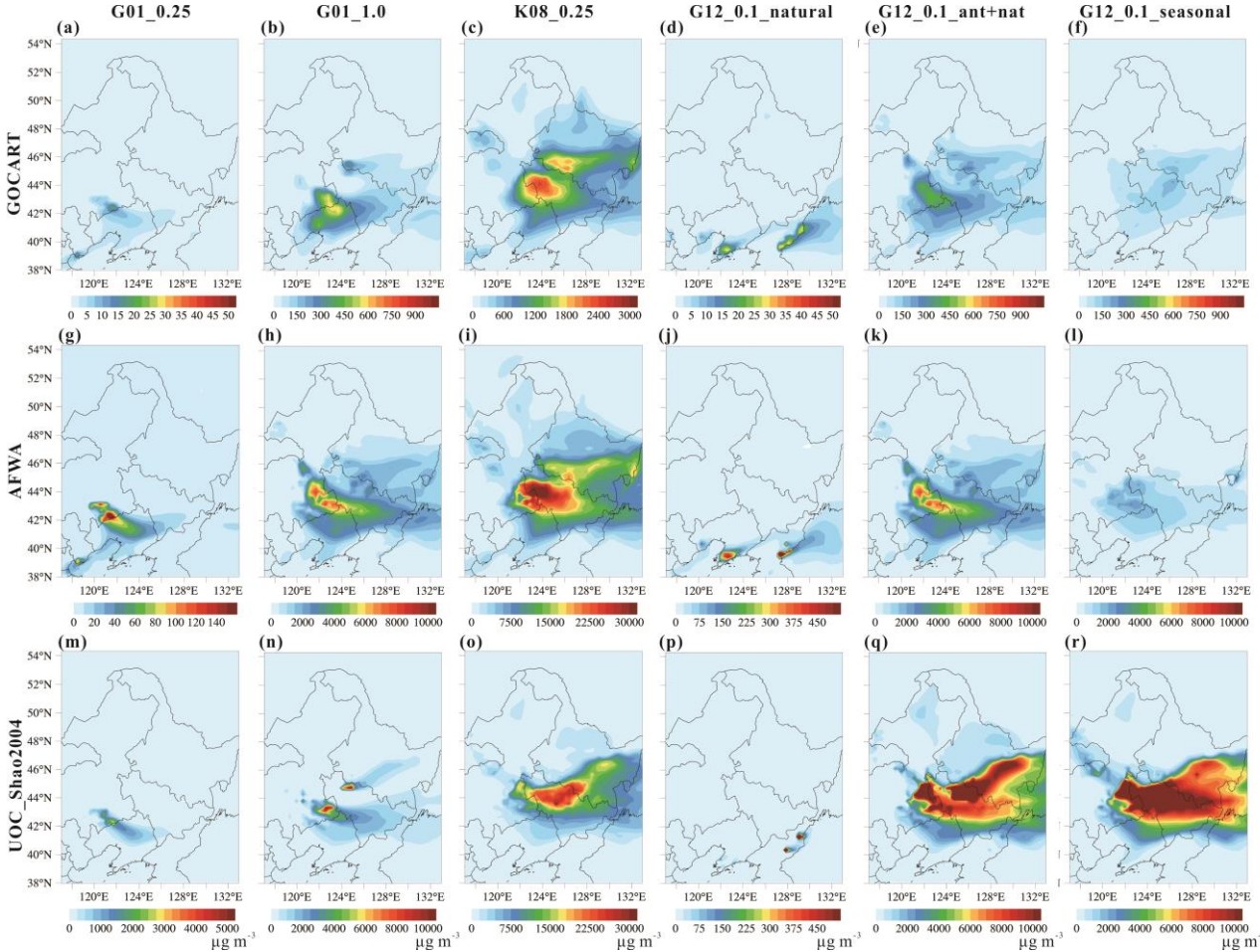

**Figure 5.** Daily mean PM$_{10}$ distributions in NEC on May 5th, 2015 using GOCART, AFWA and UOC_Shao2004 with each source map. (**a**)–(**e**): GOCART, (**g**)–(**l**): AFWA, (**m**)–(**r**): UOC_Shao2004.

## 3.3 Performance of CHIMERE v2017r4 dust simulation

The daily PM$_{10}$ patterns simulated by CHIMERE v2017r4 with different dust schemes (AGO and KOK) and three erodible fraction algorithms are illustrated in Fig. 6. The distributions of AGO PM$_{10}$ with three kinds of erodible fractions present similar patterns: two regions of higher PM$_{10}$ concentration were seen in Horqin sandy land and Wulagai Gobi, respectively. The simulated dust showed its impact on the eastern areas like Jilin and northern Liaoning Province (Fig. 6a~c),

while northeastern NEC (such as eastern part of Heilongjiang Province) were also observed to be influenced by this dust episode (Fig. 2). In comparison, there was only one source location, in Wulagai Gobi, with the KOK scheme (Fig. 6d~f), and the same dust source was also presented over NEC in the global model (Fig. 2c in Kok et al., 2014b). For the simulated results using different threshold friction velocity algorithms, the area of dust source and dust intensity with SL were smaller than those with IW (Fig. 6g~i), indicating that the dust was more difficult to emit using SL. Nevertheless, the dust emissions in Wulagai Gobi were over-predicted. Observations showed relative lower $PM_{10}$ and AOD in that area (Fig. 2).

The most striking discrepancy between the model results was in their concentration level. Daily $PM_{10}$ with erodibility derived from USGS and a combination of USGS and MODIS in the source regions exceeded 1200 μg $m^{-3}$, and ranged from 100 to 800 μg $m^{-3}$ in the transported areas. In comparison, the simulated concentrations with erodibility derived from MODIS were only about half of those values. In the KOK scheme, the simulated $PM_{10}$ concentration was < 50 μg $m^{-3}$ across the whole NEC area, thereby significantly deviating from its actual value. This difference might have arisen because the KOK scheme was mainly built on fragmentation theory (dust aggregates are fragmented by saltators into smaller particles and then emitted vertically to the atmosphere), which might be more suitable for desert land and barren land with lower cohesive energy. The strong underestimation of dust emission by the KOK scheme in NEC could be explained by the large areas of cropland with mollisol and grassland, yielding dust aggregates enriched in organic matter (Fan et al., 2010) that resist fragmentation.

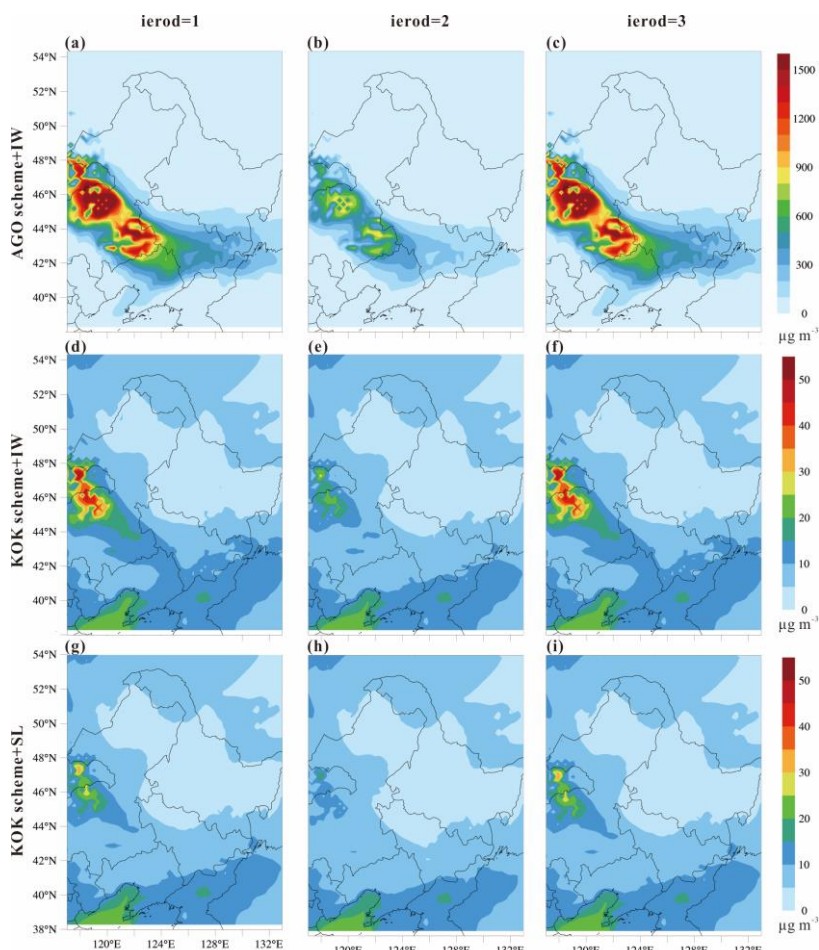

**Figure 6.** Daily mean PM$_{10}$ distributions in NEC on May 5th, 2015 using AGO (with threshold friction velocity of IW) and KOK (with threshold friction velocity of IW and SL) with erodible fractions *ierod*=1, *ierod* =2 and *ierod* =3.

### 3.4 Performance of CMAQ v5.2.1 dust simulation

5    The simulated CMAQ v5.2.1 windblown dust emission is shown in Fig. 7. The distributions with and without dust emissions from cropland both presented the same spatial pattern for daily PM$_{10}$ concentration. Dust was emitted mainly from the Horqin sandy land and a small area in western Jilin Province. Comparing to observations and simulated results of WRF-Chem and CHIMERE, the dust simulated by CMAQ was only short-distance transported southeastwards to parts of Liaoning and Jilin Province, yet had little influence on the areas of Heilongjiang Province, which could be explained as

10    having the lowest simulated dust emissions. The simulated PM$_{10}$ concentrations were about 60 (50) μg m$^{-3}$ with (without) cropland dust emission in source areas, and only ranged from 10 to 20 μg m$^{-3}$ in the transported areas. Comparing these two results, the contribution of anthropogenic wind-blown dust from cropland was only 10 μg m$^{-3}$, yet there were no further

obvious differences.

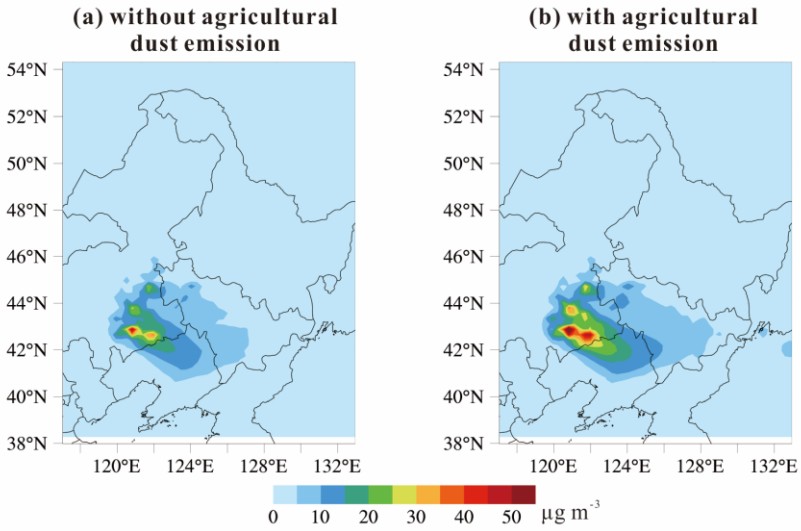

**Figure 7.** Daily mean PM$_{10}$ distributions in NEC on May 5th, 2015 without (**a**) and with (**b**) agricultural dust emission.

To determine the reason for the underestimate of dust emission flux in CMAQ, the formula and source code of the latest dust emission scheme (LS99-FENGSHA) used in CMAQ version 5.2 were analyzed. According to Equation 13 and its description in Foroutan et al. (2017), the vertical-to-horizontal dust flux ratio ($\alpha$) which determines the vertically transportable fraction of emitted dust particles is calculated via Equation 24 in Lu and Shao (1999), and parameters in this equation were defined according to Table 2 of Kang et al. (2011). The formula is expressed as follows:

$$\alpha = \frac{F}{Q} = \frac{C_\alpha g f \rho_b}{2p}(0.24 + C_\beta u_* \sqrt{\frac{\rho_p}{p}}) \tag{1}$$

where $f$ is the fraction of fine particles contained in the soil volume, $p$ is plastic pressure, in the range of $10^3 \sim 10^7$ N m$^{-2}$ (Gillett, 1977; Callebaut et al., 1985; Rice et al., 1997), $\rho_b$ and $\rho_p$ are the bulk soil and soil particle densities with unit of kg m$^{-3}$, $g$ is the gravitational constant in m s$^{-2}$, $u_*$ is friction velocity in m s$^{-1}$, and $C_\alpha$ and $C_\beta$ are constants. Here the formula described in Lu and Shao (1999) is named as LS99 and a version of LS99 modified by Kang et al. (2011) and introduced in CMAQ since version 5.2 by Foroutan et al. (2017) is called F17. The formula involving $p$ for calculating $\alpha$ according to Shao (2004), namely S04, can be described as:

$$\alpha = c_y \eta_{f,i}[(1-\gamma) + \gamma \frac{p_m(d_i)}{p_f(d_i)}]\frac{g}{u_*^2}(1 + 12u_*^2\frac{\rho_b}{p}(1 + 14u_*\sqrt{\frac{\rho_b}{p}})) \tag{2}$$

where $p_m(d_i)$ and $p_f(d_i)$ are respectively the fully and minimally disturbed dust fraction in bin $d_i$, and $\eta_{f,i}$ is the fully disturbed dust fraction. $\rho_b$=1000 kg m$^{-3}$ is bulk soil density. $\gamma$ is a function specified as $\gamma = \exp[-(u_* - u_{*t})^3]$ where $u_{*t}$ is threshold friction velocity. $c_y$ is a dimensionless coefficient which is set to be $1\times10^{-5}, 4\times10^{-5}, 5\times10^{-5}, 3\times10^{-4}$ for different soil textures

and locations in Shao (2004); then, values of soil plastic pressure $p$ in the range of $10^2$ to $10^4$ N m$^{-2}$ were obtained via matching with observed dust flux and friction velocities. This formula is now used in WRF-Chem v3.9.1. Note that the fitted $c_y$ and $p$ defined above could only be used in S04 and not in LS99 and F17 with different physical parameters. For example, the fitted value of 5000 for $p$ (silty clay loam) in Table 3 of Shao (2004) was used as $p$ of sand in Kang et al.

(2011). To correct the overestimated $p$ used in the vertical flux calculation of LS99, Kang et al. (2011) reported that a modified $C_\alpha$ was recalculated based upon $c_y$ (which is used in S04). However, to our knowledge, no method based on physical evidence is available to complete this conversion. Moreover, the source code of Shao_2004 in WRF-Chem only uses prescribed values $p = 3 \times 10^4$ and $c_y = 1 \times 10^{-5}$ without considering the soil textures. As both of their values varied widely over soil types and locations, the mismatch in part of the study domain would lead to difference in magnitude,

no matter in CMAQ or WRF-Chem.

In order to further verify the effects of modified $C_\alpha$ and $p$ used by Kang et al. (2011) and Foroutan et al. (2017) on dust vertical flux, the values of $\alpha$ for soil texture of sand, loam and clay were calculated following LS99, S04, F17 and the related formula in Marticorena and Bergametti (1995) (MB95, which was used in the original FENGSHA of CMAQ v5.0), along with measurements from laboratory experiments and field observations, as depicted in Fig. 8. Values of $\alpha$ for sandy soil

calculated via four formulae all showed better agreement with observations (Fig. 8a). For loam and sandy clay loam soil, only the result of LS99 was able to match the observed level, while those of S04 and F17 were about two orders of magnitude smaller (Fig. 8b). When considering that the dominant soil textures were loam and clay loam in NEC, this explained the reason for the underestimation occurred in CMAQ compared with WRF-Chem and CHIMERE. In addition, no $\alpha$ equation (shows in Figure S2, S3 and S5) could reproduce the observed positive correlation between $\alpha$ and friction velocity

(Fig. 8c). Furthermore, comparing to the sandblasting for the clay and clay loam, the dust originated from aerodynamic entrainment (which was not taken into account by the present dust models) was significantly constituted up to 28.3% and 146.4%, respectively (Parajuli et al., 2016). The calculations of vertical dust flux should be further examined in the future to understand its contribution to model bias for key soil types.

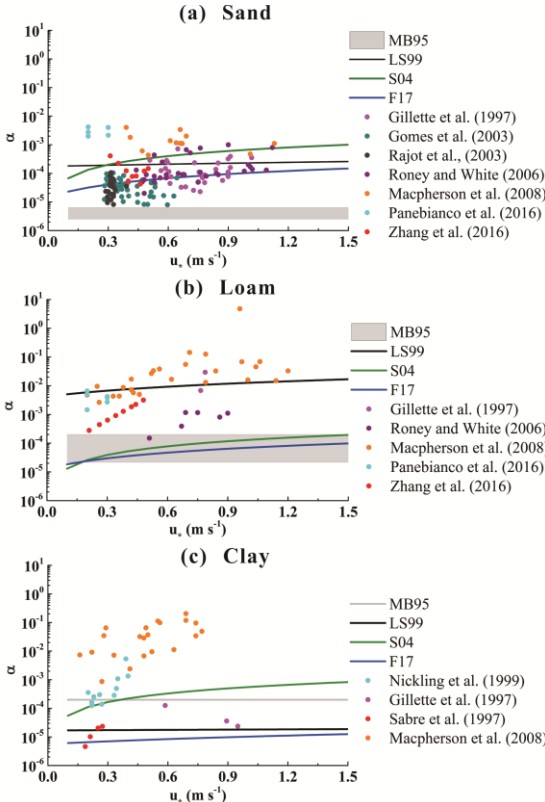

**Figure 8.** Vertical-to-horizontal dust flux ratio (α) for sand (**a**), loam (**b**) and clay (**c**) as a function of friction velocity ($u_*$) following Marticorena and Bergametti (1995) (MB95), Lu and Shao (1999) (LS99), Shao (2004) (S04) and Foroutan et al. (2017) (F17), and observations from the literature.

### 3.5  Performance of CAMx v6.50 dust simulation

As the dust mask used in CAMx showed no coverage in NEC area, the seasonal dust source map (G12_0.1_seasonal) was adapted to replace the original dust mask file as it had the best performance among those source maps in the WRF-Chem model (Fig. 5). The values in source map file were changed to 1 when the erodible fraction > 0 to fit the format of the dust mask file (Fig. 9a). Then the CAMx simulation was implemented and the daily averaged $PM_{10}$ distribution on May 5[th], 2015 is presented in Fig. 9b. It shows that the daily $PM_{10}$ concentration simulated by CAMx ranged from 0 to 30 μg m[-3] with high value in the southwest part of the simulated domain, and there was no dust emitting from any erodible area in NEC. A control simulation without dust emission was also conducted and the $PM_{10}$ pattern was same with Fig. 2b. It means that no dust emission at all and CAMx model failed to reproduce this dust episode occurred in NEC.

Considering the dust mask had been updated and the erodible areas were included in model, the poor performance of CAMx might result from the lower value of friction velocity. In the dust model of CAMx, the friction velocity is limited to a

maximum value of 0.4 m s$^{-1}$, making it keep a low level comparing to the values of other models (Fig. S11). It was difficult to exceed the $u_{*t}$ which was generally larger than 0.4 m s$^{-1}$ (Fig. S12), so no dust emission occurred. Therefore, this limitation value was subsequently removed from the source code (wbdust.f90) and the simulation was conducted again. The distribution of simulated PM$_{10}$ without the $u_*$ limitation was presented in Fig. 9c. It shows that the dust was mainly from western Jilin Province near the Songnen sandy land and transported westward. This pattern could be also observed from ground observations (Fig. 2e). However, there was no simulated dust emitting from Horqin sandy land. Simulated PM$_{10}$ concentrations were generally lower than the observations with about 120 μg m$^{-3}$ in source areas and 10~50 μg m$^{-3}$ in the transported areas. Compared with the simulation with $u_*$ limitation, this result was obviously improved which indicated that the limitation value of $u_*$ in CAMx needs further adjustment on region scale to improve its performance over the areas other than barren and sparsely vegetated area.

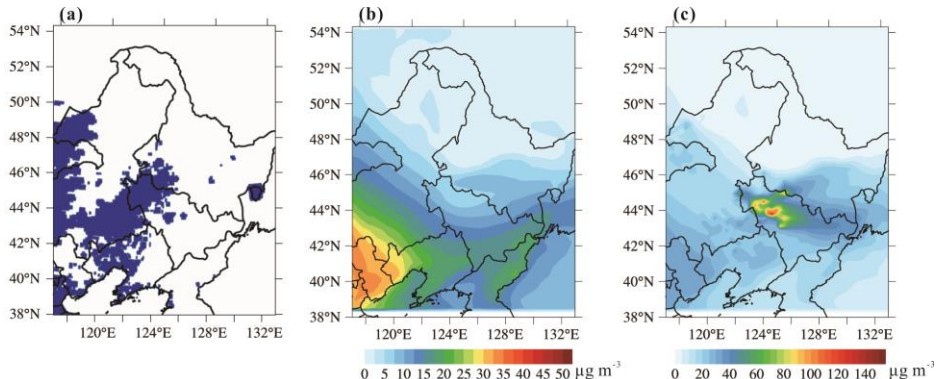

**Figure 9.** The substituted dust mask (**a**) and daily mean PM$_{10}$ distributions with (**b**) and without (**c**) friction velocity limitation in NEC on May 5th, 2015.

## 3.6 Inter-model Comparisons

The distribution and numerical values of PM$_{10}$ in each simulation were described in Sections 3.1~3.5, revealing remarkable differences between models. Most models simulated the primary dust source location (Horqin sandy land); however, many could not accurately represent other sources and the dust patterns in other parts of NEC. In this section, quantitative analyses are conducted to validate and evaluate the performances of different air quality models and dust schemes.

Considering the large discrepancies in simulated values (ranging from $10^0$ to $10^4$ μg m$^{-3}$), it is difficult to conduct the

evaluation on all of the simulations at the same time. Therefore, in this section a scaling factor is applied to the model outputs of WRF-Chem v3.9.1, CMAQ v5.2.1 and CAMx to allow a meaningful comparison against observed $PM_{10}$ concentrations. The corresponding simulations and scaling factors are summarized in Table 3. Since the patterns modeled with CHIMERE v2017r4 using different erodible fractions were quite similar, here only the outputs simulated by AGO scheme with *ierod*=3 (mixed USGS and MODIS) were chosen for further validation. Subsequently, five statistical parameters (correlation coefficient (CORR), relative mean square error (RMSE), normalized standard deviation (NSD), normalized mean bias (NMB) and normalized mean error (NME)) for the hourly data of 13 simulations and observations at 40 ground-based monitoring sites in NEC were calculated and averaged into four sub-areas of NEC (Fig. 2) for quantitative evaluation.

**Table 3.** Simulations used for validation and their corresponding tuning coefficients

| No. | Simulation | Tuning coefficient |
|-----|------------|--------------------|
| 1 | chem_gocart_g01 | 1 |
| 2 | chem_gocart_k08 | 0.5 |
| 3 | chem_gocart_g12 | 5 |
| 4 | chem_afwa_g01 | 0.25 |
| 5 | chem_afwa_k08 | 0.04 |
| 6 | chem_afwa_g12 | 0.5 |
| 7 | chem_s04_g01 | 0.5 |
| 8 | chem_s04_k08 | 0.05 |
| 9 | chem_s04_g12 | 0.07 |
| 10 | chim_ierod3 | 1 |
| 11 | cmaq | 70 |
| 12 | cmaq_agland | 70 |
| 13 | camx | 10 |

Note: 'chem' indicates WRF-Chem, 'chim' indicates CHIMERE, 'cmaq' indicates CMAQ and 'camx' indicates CAMx. 'gocart', 'afwa' and 's11' indicate the GOCART, AFWA and UOC_Shao2004 schemes, respectively. 'g01', 'k08' and 'g12' indicate the source maps of 'G01_1.0', 'K08_0.25' and 'G12_0.1_seasonal'. 'agland' indicates that the agricultural dust emission was included in CMAQ.

A Taylor diagram (Taylor, 2001) comparing in-situ observations against simulated concentrations in CTA, SWA and NEA is shown in Fig. 10. In this diagram, the NSD (ordinate) and CORR indicated each model's ability to reproduce $PM_{10}$ variability, while the RMSE (distance to point OBS) measures differences between the modeled and observed $PM_{10}$ within the three sub-areas. WRF-Chem GOCART (labeled 1~3 in Fig. 10) yielded CORR values generally below 0.5 in three sub-areas and differed greatly between different dust source maps, while CORR values with the AFWA scheme (labeled 4~6)

indicated stronger correlations in the CTA and NEA areas with values of 0.52~0.79 compared with only 0.17~0.46 in SWA. UOC_Shao2004 yielded the highest CORR values, of up to 0.82, among the four dust schemes in WRF-Chem, and the UOC_Shao2004 simulation with dust source map G12_0.1_seasonal showed the strongest correlation of all. CHIMERE and CMAQ yielded CORR values ranging from 0.43 to 0.76, with good correlations in all three areas. The CORR value of CAMx was similar with those of CHIMERE and CMAQ in NEA, but much lower in SWA (0.18).

The RMSEs and NSDs depended not only on the individual dust source maps, but also, strongly, on the tuning coefficients. Although the CORRs of WRF-Chem with GOCART were the lowest among all schemes, that combination yielded very low NSDs and RMSEs, showing that simulated concentrations were closer to the measurements. AFWA yielded relatively low NSDs and RMSEs in CTA and NEA, but the highest values in sub-area SWA. UOC_Shao2004 in CTA and NEA yielded the highest deviations. The NMBs and NMEs of the WRF-Chem simulations were lower in the CTA and SWA sub-areas than in the other two sub-areas (Fig. 11a~b). CHIMERE yielded the lowest NMB (near zero) and an NME <75%, while the NMB and NME for CMAQ were slightly larger. For CAMx, it had smaller bias and error in CTA and NEA while its NMB and NME were larger than CHIMERE and CMAQ in SWA.

In NWA, which is located in the upwind part of the study area, most simulations considerably overestimated dust: NSDs were 30~90 and CORR values were low (generally < 0.5 and < 0.2) in half the simulations. Thus, NWA was not included in the Taylor diagram. More detailed statistics are provided in Table S8.

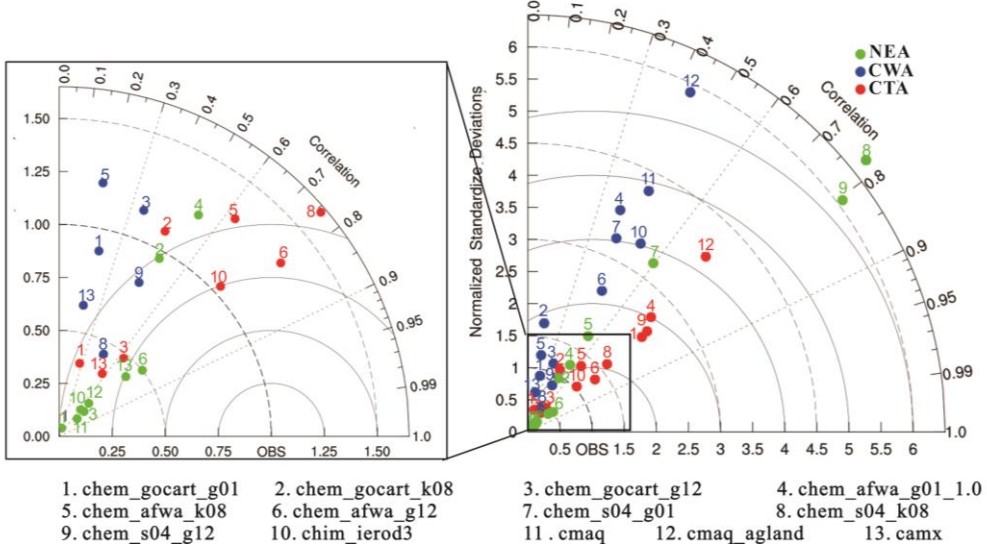

**Figure 10.** Taylor diagram comparing the hourly PM$_{10}$ concentrations of simulations with in-situ measurements for 3 regions of NEC described in Figure 2 (CTA, red circles; CWA, blue; NEA, purple). The numbers correspond to individual simulations summarized in Table 3.

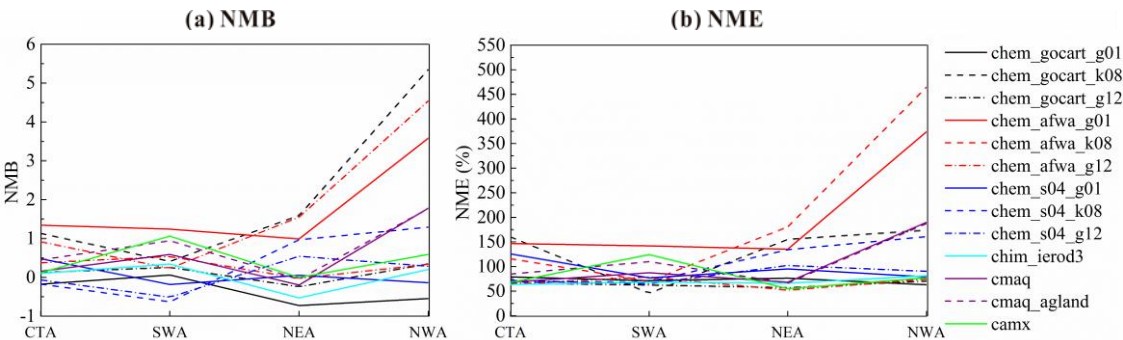

**Figure 11.** Normalized mean simulation bias (**a**) and error (**b**) relative to measurements of $PM_{10}$ during the dust period in the four regions of NEC shown in Fig. 2 (these are CTA, SWA, NEA and NWA). Simulation labels follow Figure 9.

To further illustrate the ability of each model to reproduce the temporal patterns of regional $PM_{10}$, time series of simulated hourly $PM_{10}$ concentrations and in-situ measurements at four sites in CTA, and at one site in each of NEA, SWA and NWA, are shown in Fig. 12. The models were able to reproduce the peak and high value period during May 5[th] in CTA and NEA (Fig. 12a~e); however, the simulations generally prolonged the period of high PM while often underestimating PM at sites with high dust intensities (e.g. Tongliao) and overestimating PM at lower-intensity sites such as Harbin, Shengyang and Jiamusi. One feature of note is that the $PM_{10}$ concentrations simulated by UOC_Shao2004 were still overestimated at CTA sites even after parameter tuning (Fig. 12a~12d). CHIMERE underestimated the peaks at most sites and even yielded almost steady levels at Jiamusi, whereas observations showed a moderate dust peak (Fig. 12e). Figure 12f showed that the simulations poorly represented $PM_{10}$ concentrations Jinzhou in SWA: despite showing two $PM_{10}$ peaks on this high-dust day, with strong overestimates and earlier peak timing among the different air quality models. At Hulun Buir in NWA, there was no apparent similarity with measurements among any of the models except CAMx which presented relatively close fluctuating pattern to the observation (Fig. 12g).

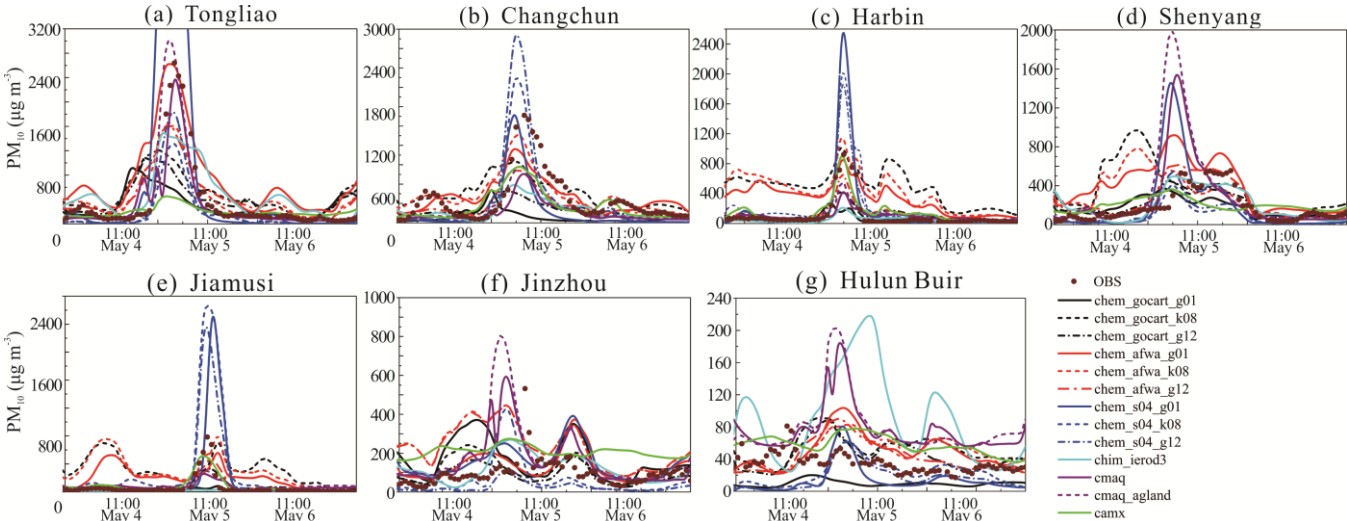

**Figure 12.** Time series (UTC) of hourly PM$_{10}$ for the simulations and measurements at 4 central sites (Tongliao (**a**), Changchun (**b**), Harbin (**c**) and Shenyang (**d**)); 1 NE site, Jiamusi (**e**); 1 SW site, Jinzhou (**f**); and 1 NW site, Hulun Buir (**g**) between May 4th and May 6th, 2015. Simulation labels follow Figure 10.

The inter-model comparisons showed that WRF-Chem GOCART underestimated dust levels and yielded the lowest correlation with measurements among all the simulations. AFWA and UOC_Shao2004 performed well in high dust regions and with different source maps but PM$_{10}$ concentrations were strongly overestimated. Different schemes in WRF-Chem presented correlations in NEC that are comparable with results in other areas, such as East Asia (Su and Fung, 2015), West Asia (Nabavi et al., 2017) and the Mediterranean (Flaounas et al., 2017; Rizza et al., 2017); however, most simulations by WRF-Chem displayed a considerable bias towards higher concentrations. In addition, the newly found source code errors after these simulations, such as bugs in GOCART gravitational settling (module_gocart_settling.F) and optical_prep_gocart routine in WRF-Chem v3.9.1 would also lead to the increase of bias and error.

The statistics for CHIMERE demonstrated its excellent performance in simulating the selected dust episode, with the lowest regional mean bias and error, as well as temporal variations that closely matched the observations (especially in southwestern NEC). Despite the generally good performance, two problems remain. First, the location of dust emissions, as dust emitted from Wulagai Gobi was not observed at the nearby monitoring sites and resulted in the discrepancy with measurements in western NEC. Second, the underestimated PM$_{10}$ in central NEC may have arisen from the omission of sub-regional dust sources or an underestimation of wind speed in central NEC (Fig. S7a) resulting in a lower friction velocity that was less likely to surpass the threshold value. Meanwhile, erroneous unit conversion of soil moisture from volumetric to mass percentage (just multiplies a value of 100 from unit of m$^3$ m$^{-3}$ to kg kg$^{-1}$ instead of via the equation showed in Figure S2) when calculating $u_{*t}$ in the CHIMERE source code would also have caused a relatively high threshold to influence the dust emission. Moreover, different algorithms for $u_{*t}$ resulted in significant differences in the simulated dust emission. For instance, the variations of $u_*$ in CHIMERE and WRF-Chem are similar (Fig. S11a, b) during the dust episode, however, $u_{*t}$

presented large discrepancies between another (Fig. S12). The value of $u_{*t}$ in CHIMERE is in the upper level of the six kinds of $u_{*t}$ used in the dust emission models while that of AFWA is much lower. This could also be one of the reasons for the overestimation in WRF-Chem AFWA. Besides the relationship between $u_{*t}$ and $u_*$, the effect of roughness length should also be considered. The high-resolution roughness length data used in this model were from the GARLAP (Global Aeolian Roughness Lengths from ASCAT and PARASOL) dataset and derived from lidar and satellite observations; their relatively low values when compared to those obtained from land use data caused the simulated dust flux to be generally lower than that obtained with land use roughness length (Menut et al., 2013). In addition, different equations were used to calculate the kinetic energy when $u_{*t} < 0.27$ m s$^{-1}$ and $0.27$ m s$^{-1} < u_{*t} < 0.55$ m s$^{-1}$ respectively (Alfaro et al., 1997), however, it was not exactly calculated according to this method in CHIMERE which might lead to deviations of the results when $u_* > 0.27$ m s$^{-1}$.

Simulations by CMAQ and CHIMERE yielded comparable correlations with observations; however, the largest tuning coefficient (with a value of 70) showed that CMAQ would seriously underestimate dust levels if used without this adjustment. Besides the reasons suggested in Section 3.4, the treatment of $u_{*t}$ as a constant in the original CMAQ FENGSHA and the algorithm of Shao and Lu (2000) in CMAQ LS99-FENGSHA (Table S9) would also result in significant differences when calculating horizontal and vertical dust fluxes. In addition, using the dynamic roughness length term when calculating $u_*$ in CMAQ led to lower $u_*$ (with mean value of 0.39 m s$^{-1}$, Fig. S11c) comparing to those in WRF-Chem and CHIMERE (0.58 m s$^{-1}$). This would be another reason for its underestimated results.

The dust simulating performance of CAMx with substituted dust mask and no $u_*$ limitation considerably improved comparing to its default configuration. Its correlations and errors to the observations could keep up with the results of CHIMERE and CMAQ, however, failure in reproducing the dust emission in Horqin sandy land accounted for the bad performance over SWA.

Finally, various uncertainties were demonstrated in the physically-based dust emission schemes, and a better understanding of these would aid future model development and enhance the accuracy of dust predictions. First of all, the dust emission research has diverged into several paths and many individual algorithms based on field observations in specific areas have been developed for particular sectors. Combining these algorithms to form a complete dust emission scheme for application in other regions would introduce unavoidable uncertainty. Several inputs are required in the dust schemes; in general, these are wind speed, precipitation, land surface characteristics (e.g. land used type, soil texture, soil moisture, surface roughness, dust source map, vegetation and snow cover), such that the accuracy and quality of external input data and meteorological models cause uncertainties to accumulate in the model prediction. Note that further validation of the meteorological simulating results and the influence of atmospheric conditions on dust emission need to be conducted in the future study. Furthermore, the choices of coefficient/constant values (such as $u_{*t}$), which are based on the familiarity and experience of the users in the study domain, introduce additional uncertainty. In addition, there is uncertainty related to the spatial resolution of air quality models at regional and global scales (Foroutan et al., 2017). Most importantly, recent studies

have suggested that dust emissions by direct aerodynamic entrainment (or convective turbulent dust emission) are about one third of sandblasting dust emissions (Li et al., 2014; Parajuli et al., 2016; Ju et al., 2018), yet the parameterizing of direct aerodynamic entrainment processes has not been implemented in the dust schemes of air quality models. This should be addressed in future work.

## 4   Summary and conclusion

In this study, we quantitatively evaluated the performance of several physically-based dust emission models in the WRF-Chem v3.9.1, CHIMERE v2017r4, CMAQ v5.2.1 and CAMx v6.50 air quality models to simulate a dust event in Northeastern China. Four dust schemes and four additionally-introduced dust source maps in WRF-Chem v3.9.1, two schemes in CHIMERE and CMAQ, and one scheme in CAMx were tested for this dust event during May 4$^{th}$~6$^{th}$, 2015. For simulations with high overestimates or underestimates, scaling factors were introduced to minimize the bias between the model and observations. Northeastern China was divided into four sub-areas for further quantitative comparisons of the simulations and observed $PM_{10}$ concentrations. These models used either dust source/mask map (WRF-Chem and CAMx), or erodible fraction (CHIMERE and CMAQ) to determine whether dust is emitted from a grid cell. Different algorithms for threshold friction velocity resulted in significant differences in the simulated dust concentration and spatial distribution. Converting the observed wind speed to near-surface friction velocity was one of the most important sources of simulated uncertainties. We demonstrated that a more accurate ratio of horizontal-to-vertical dust flux for each soil texture should be obtained in future field works.

Our evaluation revealed that the $PM_{10}$ simulated by each dust scheme in WRF-Chem yielded similar spatial patterns. AFWA and UOC_Shao2004 yielded higher correlations than GOCART, but both considerably overestimated surface dust concentrations. Of the dust source maps applied in WRF-Chem, the default G01_0.25 was incapable of reproducing the dust patterns, indicating it was not applicable in Northeastern China. The five newly-introduced dust source maps displayed better performance, and the simulations with G12_0.1_seasonal presented the best relationships with ground-based observations. CHIMERE AGO and CMAQ LS99-FENGSHA were able to reproduce the spatial distribution of the dust plume, but with incorrect dust emission sources (such as Wulagai Gobi in CHIMERE) and strong underestimates in CMAQ. All simulations performed best near the dust source areas and degraded in accuracy with downstream advection. Statistical parameters indicated that the strengths of model performances decreased in the order CTA>NEA>SWA>NWA.

In general, if the numerical tuning was included, WRF-Chem AFWA with dust source map G12_0.1_seasonal yielded the best performance among all the simulations, followed by WRF-Chem UOC_Shao2004, CHIMERE AGO, CMAQ and

WRF-Chem GOCART. Without tuning the concentration, only CHIMERE demonstrated significant correlation with relatively low bias and error, but still presented problems such as the misplaced dust sources and notable underestimates. The tuning coefficient for the outputs of WRF-Chem and CMAQ indicate the dust flux needs to be scaled during the calculation to improve their performance. A dust mask including dust emissions from regions not classified as "barren or sparsely vegetated" in CAMx should be developed by refining the land cover mask in future works and the algorithm of friction velocity needs adjustment and improvement as well. Source code errors in the air quality models need to be further debugged. In addition, a physically-based direct aerodynamic dust entrainment scheme or empirical parameterization should be implemented in the models to enhance the regional air quality forecast ability for particulates.

*Code availability.* WRF-Chem is an open-source community model. The source code is available at http://www2.mmm.ucar.edu/wrf/users/download/get_source.html. The source code of CHIMERE 2017a along with the corresponding technical documentation can be obtained from the CHIMERE web site at http://www.lmd.polytechnique.fr/chimere/. CMAQ model documentation and released versions of the source code are available on the US EPA modeling site https://www.cmascenter.org/. The source code of CAMx model is available at http://www.camx.com/. All related source codes with modified dust emission files, configuration information and the namelist files for four air quality models, and even the pre- and post-processing scripts used for this study are available online via ZENODO (https://doi.org/10.5281/zenodo.3376774).

*Author contributions.* MS, XZ and CG performed the majority of the source code reconfiguration of WRF-Chem, CHIMERE CMAQ and CAMx, and initially designed the numerical simulations to carry them out. DQT, AX, WG and CX provided help for the simulation designation. LH provided support for conducting the CAMx model. HZ and SZ provided advices on the selection and usage of observational data.MS, XZ and DQT led the analysis of the simulations, and SIE, XW, XL and MD provided professional advices. SM and XZ wrote the paper and all authors read, revised, and approved the final manuscript.

*Competing interests.* The authors declare that we have no conflict of interests.

*Acknowledgements.* This work was financially supported by the National Natural Science Foundation of China (NSFC) (No. 41571063 and 41771071), National key R&D Plan of China (No. 2017YFC0212304) and Hundred-Talent Program (Chinese Academy of Sciences, No. Y8H1021001).The authors are grateful to the website of national air quality history database for collecting and maintaining the ground-based air quality data, and appreciate NASA for providing the MODIS and CALIPSO datasets, as well as for the datasets maintenance and availability. We also thank to the two anonymous referee reviewers and

editors of Samuel Remy and David Ham for their value comments.

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
