# Peer review of "Multi-model simulations of a springtime dust storm over Northeastern China: Implications of an evaluation of four commonly used air quality models (CMAQ v5.2.1, CAMx v6.50, CHIMERE v2017r4, and WRF-Chem v3.9.1)"

_Geoscientific Model Development, 2019_

## Short Comment (SC1) · 27 Apr 2019

Hello! I would recommend to use the latest WRF-Chem version (4.1 and above), since before this version there were errors in the GOCART gravitational settling (module module_gocart_settling.F) and in optical_prep_gocart routine. For details, see here https://github.com/wrf-model/WRF/releases.

---

## Short Comment (SC2) · 7 May 2019

Hello, I enjoyed looking over this paper and I think there are some pretty interesting results. I would like to caution the authors that the UoC bug fix discussed on lines 18 through 19 in section 2.4.1 was not implemented into WRF-Chem until version 4.0. Therefore unless, the authors are using a modified version of WRF 3.9.1, the UoC bug is active in the WRF version described in the paper. If the authors are using a modified

version, that needs to be stated. Otherwise, I encourage the authors to fix this bug in their copy of WRF 3.9.1 and rerun the UoC simulations. It's an easy bug to fix: located in subroutine qwhite within the module_qf03.F in the chem directory within WRFV3. This will be substantially less burdensom than rerunning all simulations with a newer version of WRF. Additionally, I recommend that the authors refine their text in section 2.4.1 such that it is clear to the reader that the bug is not corrected in the publicly available versions of WRF-Chem until version 4.0 to avoid any confusion down the line.

---

## Short Comment (SC3) · 24 May 2019

I am writing as an executive editor of GMD to highlight some issues with the code availability section which needs to be remedied in the revised manuscript.

Thank you for the significant effort you have gone to to document the data used and the model configurations in the manuscript. Nonetheless, there are code and data availability issues that need to be addressed in the revised manuscript.

**1   Persistent, public, archives for the exact version of code used**

The code availability section points to model websites. These are not persistent archive locations and create a very high risk that future readers of your manuscript will not be able to find the code you used. It is also not clear whether any source code was modified from that downloaded. For these reasons, you should produce persistent, public archives of the exact versions of the code you used. Many authors find Zenodo is a good choice for this (https://zenodo.org).

**2   Experiment specification is incomplete**

While section 2 is quite thorough, it's simply not possible to precisely specify every single model setting and everything you did to pre- and post-process data in the manuscript. For this reason, please provide a public, persistent archive (e.g. Zenodo) of the model configuration scripts or files, and pre- and post- processing code you used. Please also ensure that the use of external data is sufficiently well documented that a future reader can obtain the same forcing data and reproduce your experiments.

---

## Referee Comment (RC1) · Anonymous Referee #1 · 10 Jun 2019

The authors describe and evaluate multiple dust simulation options available in WRF-Chem, CMAQ, CAMx, and CHIMERE using a springtime dust event in Northeastern China for comparison. The objective of the paper is to document model performance and to evaluate the sensitivity of the simulation results to key aspects of the dust emission schemes. While this discussion is useful, I find the authors have drawn conclusions about governing factors without sufficient evidence. This study attempts to assess multiple dust simulation capabilities embedded in multiple models while also

assessing the influence of dust source treatments. The paper is overall well written and organized, but there are too many degrees of freedom left unexplored to determine causality of the resultant dust concentration simulations.

I recommend revision of the manuscript, considering the following comments:

Major comments: The methodology, discussion, and results sections of this manuscript primarily focus on differences between the dust emission treatments used in each model simulation; however, the individual model descriptions provided in section 2.4 provide little to no information about the algorithms comprising these schemes. There really needs to be a succinct summary of the dust emission scheme equations discussed in this paper, either directly in the text or in the appendix section. Suggest using a model flow chart similar to the approach used in Darmenova et al. (2009) or LeGrand et al. (2019) for each dust emission scheme discussed and a symbology table.

The authors state that WRF v3.9.1 was used to generate the meteorological fields used to force all of the dust models discussed in the manuscript. This is confusing. WRF-Chem is an inline model. The dust emissions and airborne concentrations evolve simultaneously with the atmospheric conditions. In other words, the dust modules in the WRF-Chem assessments were likely subject to different environmental forcing conditions than those in the CMAQ, CHIMERE, and CAMx dust modules. Did the authors use the coupled WRF-CMAQ implementation as well? What was the output frequency of the WRF v.3.9.1 output (wrfout) files? This could potentially have significant influence on the results. Furthermore, are the CHIMERE and CMAQ dust modules configured to ingest windspeed (U) or friction velocity (u*)? The dust emission calculations described in this paper, with the exception of the WRF-Chem GOCART dust emission scheme, are calculated in terms of u*. Are the u* fields being ingested by the dust emission flux equations in WRF-Chem, CMAQ, and CHIMERE identical? If so, please add a figure showing the surface U and u* fields for a few time periods in the case study sequence. If not, please add a figure showing how they vary (especially if each model is doing its

own U to u* conversion) as this could be important for deciphering causative factors in model output discrepancies.

P13L4-7: The authors did not include the CAMx dust simulation in their in-depth analyses because the dust mask field required by the CAMx dust emission scheme did not include an erodible area in their region of interest. I'm confused by this reasoning. The dust mask and the dust source maps discussed for the other schemes in WRF-Chem (Figure 3) essentially serve the same purpose. Why test out different dust source fields in WRF-Chem but not the CAMx model? Claiming the paper includes an assessment of the CAMx model seems misleading to me. Recommend the authors either test the CAMx dust emission scheme with alternate dust source treatments similar to the exercise done for WRF-Chem, or remove the CAMx model and its discussion from the manuscript entirely.

P15L20-28: The strong dust emission magnitude from UoC and AFWA compared to GOCART in this study is somewhat unexpected given the findings discussed in the LeGrand et al. (2019) paper cited here. I don't think there's enough evidence to associate the excessive flux from the AFWA scheme with the saltation bin settings. I suspect these results may actually be related to the authors' use of the Pleim-Xiu (PX) land surface model (LSM) and Pleim (ACM2) planetary boundary layer (PBL) scheme. The U/u* conversion in the PX/ACM2 setting typically produces stronger u* values than NOAH LSM/PBL combos for equivalent U values. Operational agencies that use the AFWA dust emission scheme with the PX LSM frequently make use of the ustune tuning factor in the WRF-Chem configuration file to tone down u* values ingested by the scheme for this very reason. It would be interesting to see a time series plot of model estimated u* added to the time series plot in the appendix. If there is a strong sensitivity of dust emission scheme performance to LSM choice, it would be worth highlighting. Most other dust emission scheme assessment papers use the RUC or NOAH LSM.

The authors attribute over/under prediction of simulated dust conditions to dust emission scheme setting, but these conclusions are primarily based on comparison to daily

average PM10 distributions. Simulated PM10 errors could also be due to issues with the atmospheric conditions (e.g., vertical mixing) and/or deposition/removal treatment. The validation methodology used for this study shows daily PM10 estimates are sensitive to the dust emission scheme configuration but does not provide enough evidence to confirm causality. This is especially important to note here given that multiple model frameworks are being used for this analysis.

Section 3.5: I don't understand the rational for scaling PM10 concentrations in the inter-model comparisons. Why scale the simulation output rather than the scaling the emission fluxes?

P26L13-14: The authors claim different algorithms for threshold friction velocity (FVT) resulted in significant differences in the simulated dust concentration and spatial distribution. This finding hasn't been demonstrated in this paper. The FVT treatments associated with each model haven't been introduced (again, need for model algorithm summary to guide discussion/conclusions). Recommend adding a figure of panel plots during the peak emission period showing simulated FVT estimates for a given grain size for each dust emission scheme - or - panel plots showing u*-FVT (U - FVT in the case of GOCART).

Minor comments: P4L4-6: I would not qualify this paper as the first comprehensive evaluation of dust models for East Asia. A single dust event case study is good for examination and discussion of how the dust models function under a given forcing condition, but an extended study period with several events would be needed to truly assess model performance.

P8L10: The AFWA scheme is adapted from the dust emission scheme originally described by Marticorena and Bergametti (1995), not GOCART. It would be appropriate to cite the LeGrand et al. (2019) paper here.

P8L19: The UoC coding error was not corrected in the public code distribution until the release of WRF-Chem v4.0. It is unclear here whether or not the authors manually

corrected the coding error in their compilation of WRF-Chem v3.9.1. This was also mentioned by another community member on the forum.

P12L1: The G01 acronym hasn't been defined yet.

P12: The dust source map discussion is difficult to follow. Table 2 provides a good summary, but the labels (e.g., G01, K08, etc.) need to be introduced in the text.

P12L20-21: The authors mention they conducted a ground survey to assess representativeness of the erodibility field. What method was used for the ground survey? Is this something that was done subjectively or with in situ measurements (e.g., wind tunnel or PI-SWERL device)?

P12L20: Please list the NU-WRF version number used to acquire these fields.

Figure 3: Was the G01 1-deg field generated manually for this analysis or acquired from somewhere? The 1-deg field was replaced by the .25-deg field in the WRF-Chem repository back in 2012 and is no longer part of the standard WRF-Chem static dataset download. If it was generated manually, what process was used set the vegetation mask?

P14L1: Please describe the BELD3 dataset.

P14L3: USGS and MODIS land use datasets are brought up here with no context. Given how often land use datasets are brought up in the discussion section, the authors may want to consider listing which land use dataset was used to configure each of the model frameworks. Please also list the number of classes associated a particular dataset, since there are multiple versions of both the USGS and MODIS IGBP land use datasets.

Sections 2.4 and 2.5 in general: The overall model descriptions are somewhat vague. Additional model configuration information would be needed if others wished to replicate this study or the authors' methodology. Given the number of model frameworks used and the current paper length, it may be better to include specific model configuration setting information in a supplement. This was also noted in the review forum by GMD editor David Ham.

P15L9: "Control" is too strong of a statement here. Suggest changing to distribution and intensity of modeled dust are sensitive to...

P16L8-10: This seems like an odd choice to me. Figures 5k&5L and 5Q&5R suggest these two treatments produce markedly different results.

P20L16-19: Dust concentration could be due to other factors outside of dust emissions (e.g., dispersion, mixing, deposition treatment, etc.). The causality statement here is too strong.

P2016-21: This discussion needs equations. See previous comment about emission scheme flow charts.

P24L1-11: Were these values tuned as well?

P25L7-8: These equations need to be provided. Is this error unique to this version of CMAQ? How is soil moisture integrated into the calculation of friction velocity threshold?

P26L6: Authors evaluated dust models, not dust emission schemes. There are too many free variables to isolate result outcomes to the dust emission schemes. Use of dust models here instead of dust emission schemes would make the language here consistent with the intro section.

P26L7: "four newly-introduced dust source maps in WRF-Chem" is a bit of an overstatement. The NU-WRF model some of these maps were obtained from is the NASA implementation of WRF-Chem.

P26L14-15: I agree with the authors that uncertainty associated with the U to u* conversion is likely an important source of error, potentially more so than minor differences in dust emission physics, but there are no discussions, figures or values presented in

this paper supporting this statement.

P26L24-25: Suggest changing to "All simulations performed best near the dust source areas and degraded in accuracy with downstream advection." This may indicate issues with transport, deposition, or forcing conditions if this feature is consistent across all model configurations. Were the WRF 3.9.1 meteorological fields assessed to ensure they captured the general storm evolution prior to being applied to the various dust models?

Typos: P17L15: Typo, might?

---

## Referee Comment (RC2) · Anonymous Referee #1 · 10 Jun 2019

Errors in the WRF-Chem implementation of the GOCART settling/deposition functions likely affected the authors' results. I think it would be unfair to expect the authors to redo their experiment though since the community release of the bug fix wasn't announced or made available until after they submitted their manuscript. As an alternative, perhaps the authors could note the recent bug fix in their discussion section and comment on the need for further assessment in future studies?

---

## Referee Comment (RC3) · Anonymous Referee #2 · 29 Jul 2019

The authors present an evaluation of different well established regional air quality models in combination with various implementations of dust emissions in terms of their ability to reproduce atmospheric mineral dust observations over Northeast China.

They focus on a rather small domain which is predominantly characterised by anthropogenic agriculture related mineral dust sources as opposed to natural desert dust sources. Therefore specifically the model performance regarding these non-desert

dust sources is assessed, which represent a challenging and very important aspect of the dust cycle. On a global scale, however, the emissions from that domain play a minor role as shown in Fig. 1.

Further, they focus on one single dust event with a duration of about one day. This dust event seems to be a reasonable choice as it is clearly identified as such, it is an outbreak from sources within the model domain, and it is not significantly affected by dust transport from the west which would be typical for the region but involves sources outside the model domain.

Due to the aforementioned focussing, the results of the study do not necessarily generalise to other regions or time periods and hence the general relevance of the study might be limited, but still the study reveals differences between the models and emissions and will help to make suitable choices for future model applications and improvements.

I therefore consider the manuscript, which is overall clear and well written, to be suitable for publication in GMD, after addressing the following.

Title

To better reflect the content of the article, I suggest to change the title to

"Multi-model simulations of a springtime dust storm over Northeast China: Implications of an evaluation of three commonly used air quality models (CMAQ v5.2.1, CHIMERE v2017r4, and WRF-Chem v3.9.1)"

due to the following reasons:

1. Only one dust storm is considered. 2. Referring to East Asia is misleading as only a comparably small subregion of East Asia is considered. East Asia in contrast comprises two of the earth's major deserts, Taklamakan and Gobi, which are not at all subject of the study. 3. The evaluation of CAMx is limited to identifying that practically no emissions can be produced from within the model domain due to the MODIS based

desert mask applied in the emission scheme which precludes emissions from regions that are not barren or sparsely vegetated. While this is an important conclusion, no further evaluation of CAMx is presented and thus the present title is misleading. Considering that some efforts where made to adjust the emissions of other models, it would have been interesting to see results from CAMx after expanding the mask to include other landcover types, but I understand that this might be beyond the scope of the study. In that case I recommend to simply adjust the title, to clarify in the abstract that three models are evaluated and (as before) to discuss in the main text why CAMx is not one of them.

Page 1, line 27

"to simulate dust storms in East Asia" should read "to simulate a dust storm over Northeast China", see above

Page 5, line 8

The line in Fig. 1 is hardly recognisable as being dark blue

Page 7, Fig. 1

Please label the regions with CTA, NWA, NEA and SWA. The CALIPSO path can hardly be identified as being blue as mentioned in the caption.

Page 8, line 20

"would" or "did"?

Page 10, line 11

"omitting the effect of soil moisture" should read "omitting the term supposed to account for the effect of soil moisture"

Page 10, line 13

"maximum", not "minimum"

Page 17, line 5

In Figs. 2 and 6 it is not possible to identify trajectories, not even the approximate direction of the outflow from the source regions can be identified in Fig. 2, it is therefore hard to make this comparison. Neither do I expect any difference, as the WRF wind fields should be quite realistic.

Page 17, line 11

"The most striking feature of the model results was their concentration" should read "The most striking discrepancy between the model results was in their concentration level" or similar

Page 17, line 15

"might" should be deleted

Page 18, line 7

The sentence "With further comparison ..." needs rephrasing

Page 19, Eqs. (1) and (2) and Section 3.4 in general

Please define all variables and make sure to use units (e.g. for the p limits in Eq. (2)). The discussion would benefit from some revision because it is hard to follow what is used in (a) the different models (b) the literature cited and (c) in the present study. E.g., it would help to mention both, model name and the related citation next to each other where applicable and make use of active voice.

Page 19, Eq. (2)

Unless $u^*$ is about 1, the RHS has a discontinuity at $p = 3e4$ Nm$^{-2}$. The two cases on the RHS are limiting expressions for large and small p, it seems to be problematic to apply them on adjacent p intervals, and not use the full expression for intermediate values of p. Where does this distinction of cases and the threshold of $p = 3e4$ Nm$^{-2}$

come from?

Page 21, line 13 to 15

Please mention that you use the AGO scheme

Page 20, line 19

The sentence "Furthermore, ..." needs rephrasing

Page 22, line 3

Please mention that you analyse hourly values

Page 22, line 9

"was shown" should read "is shown"

Page 22, line 10

"abscissa" should be replaced by "distance to point OBS"

Page 23, line 6

"Thus, NSD..." should read "Thus, NWA..."

Page 23, Fig. 9

The colours for NEA and CWA are hard to distinguish

Page 24, Fig. 11

It might be worth to enlarge the figure and refine the colours

Page 26, line 25

"best near" should read "perform best close" or similar

Page 27, line 3

This clearly is not related to the resolution but simply a matter of allowing emissions

from areas not classified as desert or sparsely vegetated by refining the landcover mask.

Page 27, Author contributions

Please make sure that the order of the initials of each contributor is consistent with the author names on the title page

---

## Author Comment (AC1) · 25 Aug 2019

Dear Dr. Alexander Ukhov:

We are very grateful to your valuable suggestion on our manuscript "Multi-model simulations of springtime dust storms in East Asia: Implications of an evaluation of four commonly used air quality models (CMAQ v5.2.1, CAMx v6.50, CHIMERE v2017r4, and WRF-Chem v3.9.1)".

You recommended us to use WRF-Chem version (4.1 and above) to conduct the dust simulation, as there were errors in the GOCART gravitational settling (module module_gocart_settling.F) and in optical_prep_gocart routine. We did not catch these errors during our simulation by WRF-Chem v3.9.1 and this may lead to bias and error of the simulating results. Now we have fixed these errors in our used WRF-Chem v3.9.1. The future work will be implemented under the corrected or the latest WRF-Chem version. However, the model simulations, analyses and evaluations in this study were finished before the release of WRF-Chem v4.1. We don't have enough time to rerun all the simulations during the response stage of our manuscript. But we have pointed out these source code errors which were not corrected in this study in the manuscript.

Thank you again for your helpful advice which have helped us to improve our works.

---

## Author Comment (AC2) · 25 Aug 2019

Dear Dr. Theodore Letcher:

Thanks for your helpful comment and interests on our manuscript "Multi-model simulations of springtime dust storms in East Asia: Implications of an evaluation of four commonly used air quality models (CMAQ v5.2.1, CAMx v6.50, CHIMERE v2017r4, and WRF-Chem v3.9.1)".

As registered member of WRF-Chem community, we always updated our used WRF-Chem model according to the emails of bug reports. Thus, we corrected the bug in UOC source code (module_qf03.F) before ran the simulations. However, we find that we didn't make it clearly expressed in the manuscript and it will misunderstand the readers. Now we revise the expression into "Note that the last term in the saltation flux formula in UOC source code is expressed as $(1 + (\frac{u_{*t}}{u_*})^2)$ by error in WRF-Chem before the version of 4.0. In this study, it has been changed into $(1 + \frac{u_{*t}}{u_*})^2$ in WRF-Chem version 3.9.1 according to the description in Shao et al. (2011)." in Section 2.4.1 of the manuscript. And we uploaded all the source codes with modifications and even the pre-/post-processing scripts to the Zenodo according to the suggestion of Editor David Ham in short comment 3.

Thank you again for your valuable suggestion that have helped us to improve our manuscript.

---

## Author Comment (AC3) · 25 Aug 2019

Dear Editor David Ham: We want to thank you for your advice on our manuscript "Multi-model simulations of springtime dust storms in East Asia: Implications of an evaluation of four commonly used air quality models (CMAQ v5.2.1, CAMx v6.50, CHIMERE v2017r4, and WRF-Chem v3.9.1)". According to your suggestion, we had uploaded all the model source codes with our modifications, the namelist files of each air quality

model, the source of the input data (such as the meteorological reanalysis data and erodible fraction data), as well as pre- and post- processing scripts we used to the website of Zenodo, and the link is https://doi.org/10.5281/zenodo.3376774. Thank you again for your valuable suggestion that have helped us to improve our manuscript.

---

## Author Comment (AC4) · 25 Aug 2019

The comment was uploaded in the form of a supplement:
https://www.geosci-model-dev-discuss.net/gmd-2019-57/gmd-2019-57-AC4-supplement.zip

---

## Author Comment (AC5) · 25 Aug 2019

The comment was uploaded in the form of a supplement:
https://www.geosci-model-dev-discuss.net/gmd-2019-57/gmd-2019-57-AC5-supplement.zip

---

## Author Comment (AC6) · 25 Aug 2019

Dear referee reviewer #1: Thank you for understanding of the difficulty for redo the simulations with considering the errors of the GOCART settling/deposition functions in WRF-Chem, we admit that these error would affect the simulated results. With your suggestion, as an alternative, we had noted the recent bug fix in the discussion section and comment on the needs for further assessment in future studies. Thanks

for you helpful suggestions.

---

## Author Response (AR1)

Dear referee reviewer #1:

Many thanks to your insightful comments and valuable suggestions on our manuscript "Multi-model simulations of springtime dust storms in East Asia: Implications of an evaluation of four commonly used air quality models (CMAQ v5.2.1, CAMx v6.50, CHIMERE v2017r4, and WRF-Chem v3.9.1)". After careful discussions with other co-authors, we have carefully revised our manuscript, and written this point-to-point response letter here.

The marked-up manuscript with revisions has been presented at the end of this file. The revised or added contents are listed as follows (words in red are the responses):

1. The methodology, discussion, and results sections of this manuscript primarily focus on differences between the dust emission treatments used in each model simulation; however, the individual model descriptions provided in section 2.4 provide little to no information about the algorithms comprising these schemes. There really needs to be a succinct summary of the dust emission scheme equations discussed in this paper, either directly in the text or in the appendix section. Suggest using a model flow chart similar to the approach used in Darmenova et al. (2009) or LeGrand et al. (2019) for each dust emission scheme discussed and a symbology table.

Response: We want to thank the reviewer for the constructive and insightful advice. We realize that a succinct summary of the dust emission scheme is needed in our manuscript. According to your suggestion, we further introduced the dust emission schemes (such as the algorithms of dust flux and relevant parameters) used in each air quality model in Section 2.4. The differences between the dust schemes are also described. The flow charts for all dust emission schemes including equations, relevant literature and required input parameters, as well as variable lists are provided in the Supplementary file (Fig. S1~S6 and Table S1~S6).

2. The authors state that WRF v3.9.1 was used to generate the meteorological fields used to force all of the dust models discussed in the manuscript. This is confusing. WRF-Chem is an inline model. The dust emissions and airborne concentrations evolve simultaneously with the atmospheric conditions. In other words, the dust modules in the WRF-Chem assessments were likely subject to different environmental forcing conditions than those in the CMAQ, CHIMERE, and CAMx dust modules. Did the authors use the coupled WRF-CMAQ implementation as well? What was the output frequency of the WRF v.3.9.1 output (wrfout) files? This could potentially have significant influence on the results. Furthermore, are the CHIMERE and CMAQ dust modules configured to ingest windspeed ($U$) or friction velocity ($u_*$)? The dust emission calculations described in this paper, with the exception of the WRF-Chem GOCART dust emission scheme, are calculated in terms of $u_*$. Are the $u_*$ fields being ingested by the dust emission flux equations in WRF-Chem, CMAQ, and CHIMERE identical? If so, please add a figure showing the surface $U$ and $u_*$ fields for a few time periods in the case study sequence. If not, please add a figure showing how they vary (especially if each model is doing its own $U$ to $u_*$ conversion) as this could be important for deciphering causative factors in model output discrepancies.

Response: Thanks your hard works and these valuable comments. We found that this sentence was not correctly expressed in this part of the manuscript. WRF v3.9.1 was used to generate the meteorological fields. The output was used to drive the air quality model of CHIMERE, CMAQ and CAMx. As to WRF-Chem, it is an inline model and dust emissions are calculated simultaneously with the atmospheric fields. Therefore, we cannot say that it is driven by the WRF meteorological fields. In the manuscript, it is revised as "The Weather Research and Forecasting (WRF) model version 3.9.1 was used to conduct the

meteorological simulations, then to provide the hourly meteorological output fields to drive the air quality models of CHIMERE, CMAQ and CAMx while the chemistry module of WRF (WRF-Chem) was conducted simultaneously with the meteorological fields."

We don't use the coupled WRF-CMAQ implementation in this study. The output frequency of the WRF v.3.9.1 file for these four models is one hour.

Among these 4 air quality models, the $u_*$ fields calculated by WRF is only used in WRF-Chem model for dust flux calculation while other 3 models implement $u_*$ calculation independently. In CHIMERE v2017r4 model, $u_*$ is calculated according to $u_* = \frac{\kappa u_{10}}{\ln\left(\frac{10}{z_0}\right)}$, where $u_{10}$ is wind speed at 10m, $z_0$ is roughness length and $\kappa$ is the Karman constant. In addition, the equation of Weibull distribution $\left(p(|u_{10}|) = \frac{k}{A}\left(\frac{|u_{10}|}{A}\right)^{k-1} e^{[-(\frac{|u_{10}|}{A})^k]}\right.$, where $k$ is a dimensionless shape parameter and $A$ is modeled wind speed, in meters per second.) is introduced for wind speed adjustment (Cakmur et al., 2004; Pryor et al., 2005). The friction velocity in CMAQ v5.2.1 was calculated based on an updated dynamic relation ($\frac{z_0}{h} = \begin{cases} 0.96\lambda^{1.07} & \lambda < 0.2 \\ 0.083\lambda^{-0.46} & \lambda \geq 0.2 \end{cases}$, where $h$ is height of solid element, $\lambda$ is total roughness density) to calculate the surface roughness length relevant to small-scale dust generation processes (Foroutan et al., 2017). The $u_*$ in CAMx v6.50 is calculated according to the equation described in Louis (1979) expressed as $u_*^2 = \frac{\kappa^2 u^2}{\ln(z/z_0)^2} F_m(z/z_0, Ri_B)$, where $F_m$ is a term as the function of Richardson number $Ri_B$. In addition, it also limits the minimum maximum value of the friction velocity to 0.4 m s$^{-1}$.

All the equations for $u_*$ are presented in the flow charts of supplement file and the $U_{10}$, as well as $u_*$ variations in each model during the dust episode are showed in Fig. 1 below (Fig. S11 in Supplementary Information). It reports that the $u_*$ variations of WRF-Chem and CHIMERE are quite similar with averaged value of 0.60 m s$^{-1}$. In comparison, the $u_*$ from CMAQ presents much lower with mean value of 0.41 m s$^{-1}$. It means that the introduction of a dynamic roughness length term in CMAQ results in lower friction velocity. This could be one of the reasons of the underestimation in CMAQ. As to the $u_*$ in CAMx, it is the lowest among the values with mean value of 0.34 m s$^{-1}$because of the maximum limitation of 0.4 m s$^{-1}$, so that CAMx performed no dust emissions as the $u_*$ was lower than $u_{*t}$ during the episode.

The discussion about influence of different $u_*$ is also presented in the Section of 3.6 in the revised manuscript.

[Figure]

Figure 1. Time series of U10 (a), and $u_*$ (b) from WRF-Chem, CHIMERE, CMAQ and CAMx in Changchun City during the dust episode.

References

Cakmur, R. V., R. L. Miller, and O. Torres: Incorporating the effect of small-scale circulations upon dust emission in an atmospheric general circulation model, J. Geophys. Res., 109, D07201, doi:10.1029/2003JD004067, 2004.

Foroutan, H., Young, J., Napelenok, S., Ran, L., Appel, K. W., Gilliam, R. C. and Pleim, J. E.: Development and evaluation of a physics‐based windblown dust emission scheme implemented in the CMAQ modeling system, J. Adv. Model. Earth Syst., 9(1), 585–608, doi:10.1002/2016MS000823, 2017.

Louis, J. F.: A parametric model of vertical eddy fluxes in the atmosphere. Boundary-Layer Meteorology, 17(2), 187-202, doi: 10.1007/bf00117978, 1979.

Pryor, S., J. Schoof, and R. Barthelmie: Empirical downscaling of wind speed probability distributions, J. Geophys. Res., 110, D19109, doi: 10.1029/2005JD005899, 2005.

3. P13L4-7: The authors did not include the CAMx dust simulation in their in-depth analyses because the dust mask field required by the CAMx dust emission scheme did not include an erodible area in their region of interest. I'm confused by this reasoning. The dust mask and the dust source maps discussed for the other schemes in WRF-Chem (Figure 3) essentially serve the same purpose. Why test out different dust source fields in WRF-Chem but not the CAMx model? Claiming the paper includes an assessment of the CAMx model seems misleading to me. Recommend the authors either test the CAMx dust emission scheme with alternate dust source treatments similar to the exercise done for WRF-Chem, or remove the CAMx model and its discussion from the manuscript entirely.

Response: Thank you very much for your helpful suggestion. This question was also mentioned by reviewer #2. After discussion with all authors, we had implemented the CAMx model for further simulations. For the implementation of the dust emission scheme in CAMx, we select the seasonal dust source map (G12_0.1_seasonal in the manuscript) to replace the original dust mask file as this dust source map had the best performance among those source maps in the WRF-Chem model. The values in source map file were changed to 1 when the erodible fraction > 0 to fit the format of the dust mask file. Similarly, the performance of the CAMx dust simulation during the dust episode is analyzed and evaluated, and the result is showed in Section 3.5 of the manuscript. The daily averaged $PM_{10}$ distribution on May 5th, 2015 is presented in Fig. 2b. It shows that the daily $PM_{10}$ concentration simulated by CAMx ranged from 0 to 30 $\mu g$ $m^{-3}$ with high value area in the southwest part of the simulated domain, and there was no dust emitting from any erodible area in NEC. A control simulation without dust emission was also conducted and the $PM_{10}$ pattern is same with Fig. 2b. It means that no dust emission at all and CAMx model failed to reproduce this dust episode occurred in NEC.

Considering the dust mask had been changed and the erodible areas were included in model, the failed simulation of CAMx might result from the lower value of friction velocity. In the dust model of CAMx, the friction velocity is limited to a maximum value of 0.4 m $s^{-1}$, making it keep a low level comparing to the values of other models (Fig. 1). It was difficult to exceed $u_{*t}$ which was generally larger than 0.4 m $s^{-1}$ (Fig. 5), so no dust emission occurred. Therefore, this limitation value was subsequently removed and the simulation was conducted again. The distribution of simulated $PM_{10}$ without the $u_*$ limitation was presented in Fig. 9c. It shows that the dust was mainly from western Jilin Province near the Songnen sandy land and transported westward. This pattern could be also observed from ground observations (Fig. 2e in manuscript). However, there was no dust emitting from Horqin sandy land. Simulated $PM_{10}$ concentrations were generally lower than the observations with about 120 $\mu g$ $m^{-3}$ in source areas and 10~50 $\mu g$ $m^{-3}$ in the transported areas. Compared with the simulation with $u_*$ limitation, this result was obviously improved which indicated that the limitation value of $u_*$ in CAMx needs further adjustment to improve its performance over the areas other than barren and sparsely vegetated area.

The analysis and discussion on the CAMx result were in Section 3.5 and 3.6 of the manuscript.

[Figure]

Figure 2. The substituted dusk mask (a) and daily mean $PM_{10}$ distributions in NEC on May 5th, 2015 (b) using CAMx model.

4. P15L20-28: The strong dust emission magnitude from UoC and AFWA compared to GOCART in this study is somewhat unexpected given the findings discussed in the LeGrand et al. (2019) paper cited here. I don't think there's enough evidence to associate the excessive flux from the AFWA scheme with the saltation bin settings. I suspect these results may actually be related to the authors' use of the Pleim-Xiu (PX) land surface model (LSM) and Pleim (ACM2) planetary boundary layer (PBL) scheme. The U/$u_*$ conversion in the PX/ACM2 setting typically produces stronger $u_*$ values than NOAH LSM/PBL combos for equivalent U values. Operational agencies that use the AFWA dust emission scheme with the PX LSM frequently make use of the ustune tuning factor in the WRF-Chem configuration file to tone down $u_*$ values ingested by the scheme for this very reason. It would be interesting to see a time series plot of model estimated $u_*$ added to the time series plot in the appendix. If there is a strong sensitivity of dust emission scheme performance to LSM choice, it would be worth highlighting. Most other dust emission scheme assessment papers use the RUC or NOAH LSM.

Response: Thank you very much for your helpful comment. When we were preparing the meteorological input files, we firstly evaluated the simulation performances of WRF surface windspeed between the LSM of PX and Noah. The result showed they had same correlation coefficient (0.8) and close RMSEs (1.52 m/s for Noah-MP scheme and 1.61 m/s for Pleim-Xiu scheme). These comparisons showed close results between two schemes, however, the errors of Noah scheme had larger standard deviation showing higher dispersion than PX scheme. Considering this and the previous study by Zhang et al. (2015) which showed a good performance of PX scheme in the same research area, we finally choose PX scheme.

Later, according to this comment, we further conducted the dust emission simulation with the LSM of Noah and find that the simulated dust concentrations are much lower than those derived from PX scheme (Fig. 3). We further compared $u_*$, U10 and surface soil moisture calculated via PX and Noah scheme and the temporal variations of them are provided in Fig. 3. It shows that the variation curves of $u_*$ calculated by PX and Noah scheme are quite similar. It does not present a stronger U-$u_*$ conversion in the PX/ACM2 setting than in Noah scheme over the research area at this time. By contrast, the Noah surface soil moisture shows larger difference, with values 93.6% higher in Changchun City and 29.6% higher in the NEC area (Fig. 4 and Table 1). Moreover, the soil moisture curve with two LSM schemes are quite different. These discrepancies may result in the differences of estimated dust emissions. The lower soil moisture (which makes smaller threshold friction velocity) simulated by using PX scheme could be the reason of the stronger dust emission magnitude from PX compared to Noah LSM scheme. However, the dust emission simulated by AFWA scheme is considerably higher than that by GOCART no matter which LSM is used. Therefore, we think the differences of wind velocity and soil moisture between PX and Noah scheme could lead to the dust emission discrepancies, but it is hard to explain the differences of dust emission magnitude between

GOCART and AFWA scheme.

Many researches indicate higher dust emission simulated by GOCART than that of AFWA scheme over the areas like Mediterranean, West Asia and southwest Asia (Flaounas et al., 2017; Nabavi et al., 2017; LeGrand et al., 2019) which are quite different from our study. So we try to find out the reasons for large discrepancies between different dust schemes under the same meteorological condition. The saltation bin configuration which influences the dust mass distributions described in LeGrand et al. (2019) maybe have impact on the dust emission and concentration. One another explanation of the over-prediction of the AFWA dust concentration is that AFWA scheme considers vertical dust flux only related to the clay content which results in a higher vertical-to-horizontal dust flux ratio (Kang et al., 2010; Rizza et al., 2016; Rizza et al., 2017). We have added this into Section 3.2 of our new revised manuscript. Meanwhile, we also add the discussion about the LSM influence on the dust simulation in the same part.

[Figure]

Figure 3. Daily mean $PM_{10}$ distributions in NEC on May 5th, 2015 using GOCART, AFWA and UOC_Shao2004 with LSM of Noah.

[Figure]

Figure 4. Time series of $u_*$, U10 and surface soil moisture simulated via LSM of PX and Noah in Changchun City during the dust episode.

Table 1. Mean $u_*$, U10 and surface soil moisture simulated via LSM of PX and Noah in the research area of NEC

|  | PX | NOAH |
| --- | --- | --- |
| U10 (m s$^{-1}$) | 5.10 | 4.66 |
| $u_*$ (m s$^{-1}$) | 0.51 | 0.46 |
| soil moisture (m$^3$ m$^{-3}$) | 0.27 | 0.35 |

Response: Thanks again. The $PM_{10}$ concentrations simulated by different models varies from $10^0$ to $10^4$ ug/m³ which means considerable discrepancies. The bias and error between simulated and observed $PM_{10}$ concentration differ widely as well. It is

very hard to plot all of the simulated concentrations or validation results in one figure and inconvenience to conduct the evaluation on all of the simulations at the same time. Therefore, we use scaling factor to adjust the outputs of WRF-Chem and CMAQ (as well as CAMx) to make them have similar concentration level with others. The scaling factor could help us to better understand the advantages and disadvantages of the model performance on dust simulation, rather than improve the accuracy of dust modeling at this time.

The evaluation in this study indicates tuning factor for dust flux is needed to improve model performance and puts forward an improving direction of dust simulation in present air quality models. Brief discussion and explanation about this are added in the parts of comparison and summary. The further work on tuning and localization of the dust models will be conducted basing on this study.

7. P26L13-14: The authors claim different algorithms for threshold friction velocity (FVT) resulted in significant differences in the simulated dust concentration and spatial distribution. This finding hasn't been demonstrated in this paper. The FVT treatments associated with each model haven't been introduced (again, need for model algorithm summary to guide discussion/conclusions). Recommend adding a figure of panel plots during the peak emission period showing simulated FVT estimates for a given grain size for each dust emission scheme - or - panel plots showing u*-FVT (U-FVT in the case of GOCART).

Response: Thank you very much for your helpful suggestion. We have calculated the FVT in each dust emission model according to the algorithms described in relevant references and source codes in models. The FVT variation used in each dust emission model during the dust episode is showed in Fig. 5. We have added this into Supplementary Information and also have discussion about its influence to dust emission in Section 3.6 of the manuscript.

From the figure we find that the overestimation of WRF-Chem AFWA could in part be explained by the lower FVT comparing to those used in other dust schemes. The variation of UOC FVT fluctuated widely and presented highest FVT peaks, but during the dust episode (5 May, 2015) it kept lower values making it in the middle level of FVT. This may be part of the reason of the overestimated $PM_{10}$ concentration. However, the FVT of GOCART has the lowest value while its simulated dust emission is in the lower level. As we know, the FVT is very important in the dust emission calculation, however, it needs further adjustment and improvement as the air quality at present have difficulty in calculating FVT properly in some areas such as northeastern China. And this is also our next step work to implement field study and measurement of FVT, adjust and localize the FVT algorithm and the relevant parameters.

[Figure]

Figure 5. Time series of $u_{*t}$ in from dust emission model of GOCART WRF-Chem, AFWA WRF-Chem, UOC WRF-Chem, CHIMERE, CMAQ and CAMx in Changchun City during the dust episode.

Minor comments:

1. P4L4-6: I would not qualify this paper as the first comprehensive evaluation of dust models for East Asia. A single dust event case study is good for examination and discussion of how the dust models function under a given forcing condition, but an extended study period with several events would be needed to truly assess model performance.

Response: Thank you for the helpful comment. This study is not the first comprehensive evaluation of dust models for East Asia, but we are sure it is the first evaluation of the dust modules in air quality models (rather than climate model (Uno et al., 2006), global model et al.). In order to keep in rigorous, this sentence is now revised as "Here we present a comprehensive evaluation of multi-model simulations of windblown dust emissions in air quality models during a dust episode in East Asia…" Now we are working on the extended evaluations of the dust emissions in air quality models, such as the evaluation on the performance of CHIMERE during an autumnal dust episode (Ma et al., 2019).

Response: It has revised as "This difference might have arisen because the KOK scheme was mainly built on fragmentation theory"

In addition, besides the major and minor questions as mentioned above, we conducted reproducibility tests of our simulations in this study, and found an error in the WRF-Chem section. We have run the WRF-Chem model with UOC_Shao2011 scheme and it showed that $PM_{10}$ simulated by UOC_Shao2011 presented little similarity with the observations (Fig. 6). Considering its unreasonable results, then we selected UOC_Shao2004 for the subsequently simulations. It means that the used dust scheme in original manuscript is UOC_Shao2004. However, the texts expressed the used scheme as UOC_Shao2011 by wrong. Therefore, we corrected this into UOC_Shao2004 in the manuscript.

[Figure]

Figure 6. Daily mean $PM_{10}$ distributions in NEC on May 5th, 2015 using WRF-Chem UOC_Shao2011.

We have tried our best to improve the manuscript and made changes in the manuscript. These changes will not influence the content and framework of the paper. And here we do not list the changes but marked in the revised manuscript. We thank the reviewer again for the constructive advice that have helped us to improve our manuscript.

All in a word, via these evaluation works, we hope to do some contributions to the community for enhance the dust forecast ability on regional scale in air quality models.

Dear referee reviewer #2:

We are very grateful for your constructive comments and suggestions on our manuscript "Multi-model simulations of springtime dust storms in East Asia: Implications of an evaluation of four commonly used air quality models (CMAQ v5.2.1, CAMx v6.50, CHIMERE v2017r4, and WRF-Chem v3.9.1)". After carefully discussions with other co-authors, we have revised our manuscript according to this point-to-point response letter.

The marked-up manuscript with revisions has been presented at the end of this file. The point to point responses are listed as follows:

1. Title. To better reflect the content of the article, I suggest to change the title to "Multi-model simulations of a springtime dust storm over Northeast China: Implications of an evaluation of three commonly used air quality models (CMAQ v5.2.1, CHIMERE v2017r4, and WRF-Chem v3.9.1)" due to the following reasons: 1. Only one dust storm is considered. 2. Referring to East Asia is misleading as only a comparably small subregion of East Asia is considered. East Asia in contrast comprises two of the earth's major deserts, Taklamakan and Gobi, which are not at all subject of the study. 3. The evaluation of CAMx is limited to identifying that practically no emissions can be produced from within the model domain due to the MODIS based desert mask applied in the emission scheme which precludes emissions from regions that are not barren or sparsely vegetated. While this is an important conclusion, no further evaluation of CAMx is presented and thus the present title is misleading. Considering that some efforts where made to adjust the emissions of other models, it would have been interesting to see results from CAMx after expanding the mask to include other landcover types, but I understand that this might be beyond the scope of the study. In that case I recommend to simply adjust the title, to clarify in the abstract that three models are evaluated and (as before) to discuss in the main text why CAMx is not one of them.

Response: We want to thank the reviewer for the constructive and insightful advice. According this comment and the suggestion from Reviewer 1. The further simulation and analysis of CAMx were implemented, and provided in Section 3.5 and the supplement file of the manuscript. As the dust mask used in CAMx showed no coverage in NEC area, the seasonal dust source map (G12_0.1_seasonal) was adapted to replace the original dust mask file as it had the best performance among those source maps in the WRF-Chem model. Therefore, the title is changed to "Multi-model simulations of a springtime dust storm over Northeastern China: Implications of an evaluation of four commonly used air quality models (CMAQ v5.2.1, CAMx v6.50, CHIMERE v2017r4, and WRF-Chem v3.9.1)"

2. Page 1, line 27. "to simulate dust storms in East Asia" should read "to simulate a dust storm over Northeast China", see above

Response: It is revised as "This study applies and evaluates four widely used regional air quality models to simulate dust storms in Northeastern China."

3. Page 5, line 8. The line in Fig. 1 is hardly recognizable as being dark blue.

Response: Thank you for your reminding. We find that the color blue didn't present well when converting to PDF format. So we changed it to color red and the revised figure is showed below.

[Figure]

4. Page 7, Fig. 1. Please label the regions with CTA, NWA, NEA and SWA. The CALIPSO path can hardly be identified as being blue as mentioned in the caption.

Response: The labels of subareas of CTA, NWA, NEA and SWA were added on the figure and the CALIPSO path color was changed to red. The revised figure is showed below.

[Figure]

5. Page 8, line 20. "would" or "did"?

Response: It has revised to "This revision could increase the saltation flux by a factor of 2 or more." According to the description in LeGrand et al. (2019).

Refference

LeGrand, S. L., Polashenski, C., Letcher, T. W., Creighton, G. A., Peckham, S. E., and Cetola, J. D.: The AFWA dust emission scheme for the GOCART aerosol model in WRF-Chem v3.8.1, Geosci. Model Dev., 12, 131-166, https://doi.org/10.5194/gmd-12-131-2019, 2019.

6. Page 10, line 11. "omitting the effect of soil moisture" should read "omitting the term supposed to account for the effect of soil moisture"

Response: Thank you for your very helpful suggestion. This sentence is revised as "The major modifications were omitting the term supposed to account for the effect of soil moisture on dust emission".

7. Page 10, line 13. "maximum", not "minimum"

Response: Thanks for your comment, it should be "maximum" and had been revised in our new manuscript.

8. Page 17, line 5. In Figs. 2 and 6 it is not possible to identify trajectories, not even the approximate direction of the outflow from the source regions can be identified in Fig. 2, it is therefore hard to make this comparison. Neither do I expect any difference, as the WRF wind fields should be quite realistic.

Response: Thank you very much for pointing out this problem. We find that this part could not properly expressed. According to the observations, the large areas of NEC such as northern Liaoning, Jilin and eastern Heilongjiang Province were influenced by this dust episode while the simulated results of CHIMERE did not show an obvious impact on northeastern NEC (eastern Heilongjiang Province). This is one of the differences between CHIMERE simulation and observation. It should be simulated and observed patterns rather than trajectories. Therefore, we revised this sentence to "The simulated dust showed its impact on the eastern areas like Jilin and northern Liaoning Province (Fig. 6a~c), while northeastern NEC (such as eastern part of Heilongjiang Province) were also observed to be influenced by this dust episode (Fig. 2).".

9. Page 17, line 11. "The most striking feature of the model results was their concentration" should read "The most striking discrepancy between the model results was in their concentration level" or similar.

Response: It has revised to "The most striking discrepancy between the model results was in their concentration level.".

10. Page 17, line 15. "might" should be deleted

Response: Thanks again. This is revised to "This difference might have arisen because the KOK scheme was mainly built on fragmentation theory".

11. Page 18, line 7. The sentence "With further comparison ..." needs rephrasing

Response: It is revised as "Comparing to observations and simulated results of WRF-Chem and CHIMERE, the dust simulated by CMAQ was only short-distance transported southeastwards to….".

12. Page 19, Eqs. (1) and (2) and Section 3.4 in general. Please define all variables and make sure to use units (e.g. for the $p$ limits in Eq. (2)). The discussion would benefit from some revision because it is hard to follow what is used in (a) the different models (b) the literature cited and (c) in the present study. E.g., it would help to mention both, model name and the related citation next to each other where applicable and make use of active voice.

Response: The definitions of variables and parameters used in equations were explained more detailed in the manuscript. Moreover, we also named the methods described in literature or used in models: the formula described in Lu and Shao (1999)

is named as LS99 and a version of LS99 modified by Kang et al. (2011) and introduced in CMAQ since version v5.2 by Foroutan et al. (2017) is called F17. The formula involving $p$ for calculating $\alpha$ described in Shao (2004) is named as S04. This part is now revised as "The formula is expressed as follows:

$$\alpha = \frac{F}{Q} = \frac{C_\alpha g f \rho_b}{2p}(0.24 + C_\beta u_* \sqrt{\frac{\rho_p}{p}}) \tag{1}$$

where $f$ is the fraction of fine particles contained in the soil volume, $p$ is plastic pressure, in the range of $10^3 \sim 10^7$ N m$^{-2}$ (Gillett, 1977; Callebaut et al., 1985; Rice et al., 1997), $\rho_b$ and $\rho_p$ are the bulk soil and soil particle densities with unit of kg m$^{-3}$, $g$ is the gravitational constant in m s$^{-2}$, $u_*$ is friction velocity in m s$^{-1}$, and $C_\alpha$ and $C_\beta$ are constants. Here the formula described in Lu and Shao (1999) is named as LS99 and a version of LS99 modified by Kang et al. (2011) and introduced in CMAQ since version 5.2 by Foroutan et al. (2017) is called F17. The formula involving $p$ for calculating $\alpha$ according to Shao (2004), namely S04, can be described as:

$$\alpha = c_y \eta_{f,i}[(1-\gamma) + \gamma \frac{p_m(d_i)}{p_f(d_i)}] \frac{g}{u_*^2}(1 + 12u_*^2 \frac{\rho_b}{p}(1 + 14u_* \sqrt{\frac{\rho_b}{p}})) \tag{2}$$

Where $p_m(d_i)$ and $p_f(d_i)$ are respectively the fully and minimally disturbed dust fraction in bin $d_i$, and $\eta_{f,i}$ is the fully disturbed dust fraction. $\rho_b$=1000 kg m$^{-3}$ is bulk soil density. $\gamma$ is a function specified as $\gamma = \exp[-(u_* - u_{*t})^3]$ where $u_{*t}$ is threshold friction velocity. $c_y$ is a dimensionless coefficient which is set to be $1\times10^{-5}$, $4\times10^{-5}$, $5\times10^{-5}$, $3\times10^{-4}$ for different soil textures and locations in Shao (2004); then, values of soil plastic pressure $p$ in the range of $10^2$ to $10^4$ N m$^{-2}$ were obtained via matching with observed dust flux and friction velocities. This formula is now used in WRF-Chem v3.9.1."

13. Page 19, Eq. (2). Unless $u_*$ is about 1, the RHS has a discontinuity at $p$ = 3e4 N m$^{-2}$. The two cases on the RHS are limiting expressions for large and small $p$, it seems to be problematic to apply them on adjacent $p$ intervals, and not use the full expression for intermediate values of $p$. Where does this distinction of cases and the threshold of $p$ = 3e4 N m$^{-2}$ come from?

Response: The threshold of p = 3e+5 N m$^{-2}$ (however, it was miswritten as 3e+4 N m$^{-2}$ in the manuscript) was from the description of Shao et al. (2004) "For p >3e+5 N m$^{-2}$, $\sigma_m$ becomes negligibly small (<0.1) under normal wind conditions, implying that saltation bombardment is insignificant in such circumstances and aggregates disintegration is the main mechanism for dust emission." Nevertheless, the reference doesn't clearly present the threshold value of the equation and no explanation about how the latter part of the equation ($\alpha = 168c_y[\eta_{mi} + (1-\gamma)\eta_{ci}]\left[\frac{1000}{p}\right]^{\frac{3}{2}} u_* g$) established is provided, and it has not been used in the air quality model. We read the reference again and found that equation 2 in the manuscript was improperly introduced. The equation 6 in Shao et al. (2004) which showed as

$$\alpha = c_y \eta_{f,i}[(1-\gamma) + \gamma \frac{p_m(d_i)}{p_f(d_i)}] \frac{g}{u_*^2}(1 + 12u_*^2 \frac{\rho_b}{p}(1 + 14u_* \sqrt{\frac{\rho_b}{p}}))$$

is used in WRF-Chem model and it doesn't need piecewise $p$ values for calculation. Therefore, this part was revised and the discussion about the threshold of $p$ is removed. Now it is shows as "Note that the fitted $c_y$ and $p$ defined above could only be used in S04 and not in LS99 and F17 with different physical parameters. For example, the fitted value of 5000 for $p$ (silty clay loam) in Table 3 of Shao (2004) was used as $p$ of sand in Kang et al. (2011). To correct the overestimated $p$ used in the vertical flux calculation of LS99, Kang et al. (2011) reported that a modified $C_\alpha$ was recalculated based upon $c_y$ (which is used in S04). However, to our knowledge, no method based on physical evidence is available to complete this conversion. Moreover, the source code of Shao_2004 in WRF-Chem only uses prescribed values $p = 3 \times 10^4$ and $c_y = 1 \times 10^{-5}$

without considering the soil textures. As both of their values varied widely over soil types and locations, the mismatch in part of the study domain would lead to difference in magnitude, no matter in CMAQ or WRF-Chem."

14. Page 21, line 13 to 15. Please mention that you use the AGO scheme

Response: Thank you for your helpful suggestion. It is revised as "here only the outputs simulated by AGO scheme with *ierod*=3 (mixed USGS and MODIS) were chosen for further validation."

15. Page 20, line 19. The sentence "Furthermore, ..." needs rephrasing

Response: It is rephrased as "Furthermore, comparing to the sandblasting for the clay and clay loam, the dust originated from aerodynamic entrainment (which was not taken into account by the present dust model) was significantly constituted up to 28.3% and 146.4%, respectively"

16. Page 22, line 3. Please mention that you analyse hourly values.

Response: This sentence is revised as "…and normalized mean error (NME)) for the hourly data of 12 simulations and observations at 40 ground-based monitoring sites in NEC were calculated…."

17. Page 22, line 9. "was shown" should read "is shown"

Response: It is revised to "is shown"

18. Page 22, line 10. "abscissa" should be replaced by "distance to point OBS"

Response: It has been revised to "…while the RMSE (distance to point OBS) measures differences between the modeled and observed $PM_{10}$…."

19. Page 23, line 6. "Thus, NSD..." should read "Thus, NWA..."

Response: It is revised to "Thus, NWA was not included in the Taylor diagram."

20. Page 23, Fig. 9. The colours for NEA and CWA are hard to distinguish

Response: Thank you for this helpful comment. The color of NEA was changed from purple to green and the revised figure is showed below.

[Figure]

21. Page 24, Fig. 11. It might be worth to enlarge the figure and refine the colours

Response: This figure had been enlarged and we also carefully refined the colors. It now shows as:

[Figure]

22. Page 26, line 25. "best near" should read "perform best close" or similar

Response: It is revised as "All simulations performed best near the dust source areas and degraded in accuracy…."

23. Page 27, line 3. This clearly is not related to the resolution but simply a matter of allowing emissions from areas not classified as desert or sparsely vegetated by refining the landcover mask.

Response: Thank for your very helpful suggestion. This sentence is revised as "A dust mask including dust emissions from regions not classified as "barren or sparsely vegetated" in CAMx should be developed by refining the land cover mask in future works."

24. Page 27, Author contributions. Please make sure that the order of the initials of each contributor is consistent with the author names on the title page

Response: Thank you for your valuable suggestion. The part of Author contributions is now revised as "MS, XZ and CG performed the majority of the source code reconfiguration of WRF-Chem, CHIMERE, CMAQ and CAMx, and initially designed the numerical simulations to carry them out. DQT, AX, WG and CX provided help for the simulation designation. LH provided support for conducting the CAMx model. HZ and SZ provided advices on the selection and usage of observational data. MS, XZ and DQT led the analysis of the simulations, and SIE, XW, XL and MD provided professional advices. SM and XZ wrote the paper and all authors read, revised, and approved the final manuscript."

In addition, we conducted reproducibility tests of our simulations and found an error in the WRF-Chem section. We have run the WRF-Chem model with UOC_Shao2011 scheme and the daily mean $PM_{10}$ on May 5th, 2015 simulated by UOC_Shao2011 is provided below. It showed that spatial pattern and concentration level had little similarity with the observations. Considering its unreasonable results, then we selected UOC_Shao2004 for the subsequently simulations. It means that the actually used dust scheme in manuscript is UOC_Shao2004. However, the texts expressed the used scheme as UOC_Shao2011 by wrong. Therefore, we corrected this into UOC_Shao2004 in the manuscript.

[Figure]

Thank the reviewer again for the constructive criticisms that have helped us to improve our manuscript. We have tried our best to improve the manuscript and made changes in the manuscript. These changes will not influence the content and framework of the paper. And here we do not list the changes but marked in the revised manuscript.

All in a word, via these evaluation works, we hope to do some contributions to the community for enhance the dust forecast ability on regional scale in air quality models.

[revised manuscript text omitted]